# Sleep spindle maturity promotes slow oscillation-spindle coupling across child and adolescent development

Ann-Kathrin Joechner[1]*, Michael A Hahn[2,3,4], Georg Gruber[5,6], Kerstin Hoedlmoser[2,3], Markus Werkle-Bergner[1]*

[1]Center for Lifespan Psychology, Max Planck Institute for Human Development, Berlin, Germany; [2]Department of Psychology, Laboratory for Sleep, Cognition and Consciousness Research, University of Salzburg, Salzburg, Austria; [3]Centre for Cognitive Neuroscience Salzburg (CCNS), University of Salzburg, Salzburg, Austria; [4]Hertie-Institute for Clinical Brain Research, University Medical Center Tuebingen, Tuebingen, Germany; [5]Department of Psychiatry and Psychotherapy, Medical University of Vienna, Vienna, Austria; [6]The Siesta Group, Vienna, Austria

*For correspondence:
joechner@mpib-berlin.mpg.de
(A-KJ);
werkle@mpib-berlin.mpg.de
(MW-B)

Competing interest: The authors declare that no competing interests exist.

**Abstract** The synchronization of canonical fast sleep spindle activity (12.5–16 Hz, adult-like) precisely during the slow oscillation (0.5–1 Hz) up peak is considered an essential feature of adult non-rapid eye movement sleep. However, there is little knowledge on how this well-known coalescence between slow oscillations and sleep spindles develops. Leveraging individualized detection of single events, we first provide a detailed cross-sectional characterization of age-specific patterns of slow and fast sleep spindles, slow oscillations, and their coupling in children and adolescents aged 5–6, 8–11, and 14–18 years, and an adult sample of 20- to 26-year-olds. Critically, based on this, we then investigated how spindle and slow oscillation maturity substantiate age-related differences in their precise orchestration. While the predominant type of fast spindles was development-specific in that it was still nested in a frequency range below the canonical fast spindle range for the majority of children, the well-known slow oscillation-spindle coupling pattern was evident for sleep spindles in the adult-like canonical fast spindle range in all four age groups—but notably less precise in children. To corroborate these findings, we linked personalized measures of fast spindle maturity, which indicate the similarity between the prevailing development-specific and adult-like canonical fast spindles, and slow oscillation maturity, which reflects the extent to which slow oscillations show frontal dominance, with individual slow oscillation-spindle coupling patterns. Importantly, we found that fast spindle maturity was uniquely associated with enhanced slow oscillation-spindle coupling strength and temporal precision across the four age groups. Taken together, our results suggest that the increasing ability to generate adult-like canonical fast sleep spindles actuates precise slow oscillation-spindle coupling patterns from childhood through adolescence and into young adulthood.

## Editor's evaluation

This is an important analysis of sleep datasets across different age groups that contributes to our understanding of sleep spindle and slow oscillation dynamics during development. The work is expected to be of interest to interdisciplinary fields including development and sleep. The analyses are solid and adequately complex to capture the changes in sleep spindle to slow oscillation coupling among the age groups.

**eLife digest** Cells in the brain are wired together like an electric circuit that can relay information from one area of the brain to the next. Even when sleeping, the human brain continues to send signals to process information it has encountered during the day. This results in two patterns of electrical activity that define the sleeping brain: slowly repeating waves (or slow oscillations) and rapid bursts of activity known as sleep spindles.

Although slow oscillations and sleep spindles are generated in different regions of the brain, they often happen at the same time. This syncing of activity is thought to help different parts of the brain to communicate with each other. Such communication is essential for new memories to become stable and last a long time.

In children, slow oscillations and sleep spindles appear together less frequently, suggesting that these co-occurring patterns of electrical activity develop as humans grow into adults. Here, Joechner et al. set out to understand what drives slow oscillations and sleep spindles to start happening at the same time.

The team used a technique called electroencephalography (or EEG for short) to study the brain activity of children, teenagers and adults as they slept. This revealed that slow oscillations and sleep spindles occur together less often in children compared to teenagers and adults. Moreover, the slow oscillations and sleep spindles observed in the children had very different physical characteristics to those observed in adults. Further analyses showed that the more similar the children's sleep spindles were to adult spindles, the more consistently they appeared at the same time as the slow oscillations.

The findings of Joechner et al. suggest that as children grow up, their sleep spindles become more adult-like, causing the spindles to happen at the same time as slow oscillations more consistently. This indicates that brain circuits that generate sleep spindles may play an essential role in developing successful communication networks in the human brain. In the future, this work may ultimately provide new insights into how age-related changes to the brain contribute to cognitive development, and suggests sleep as a potential intervention target for neurodevelopmental disorders.

## Introduction

The grouping of sleep spindles (9–16 Hz; *Cox et al., 2017*) into sequences of increased and decreased activity by the sleep slow oscillation (SO, 0.5–1 Hz; *Steriade, 2006*) during non-rapid eye movement sleep (NREM) has been recognized as an intrinsic property of the healthy, mature mammalian corticothalamic system for decades (*Contreras et al., 1996*; *Contreras and Steriade, 1995*; *Mulle et al., 1986*; *Staresina et al., 2015*; *Steriade et al., 1993*). The joint depolarization of large groups of cortical neurons during the SO up state impinges on neurons of the reticular thalamic nucleus, there, creating conditions that facilitate thalamic spindle generation (*Steriade, 1999*). Sleep spindles then propagate to the cortex via thalamocortical projections, where they promote synaptic plasticity through changes in calcium activity (*Niethard et al., 2018*; *Rosanova and Ulrich, 2005*). Yet, little is known about how this precise coalescence develops across childhood and adolescence.

Far from being an epiphenomenon, accumulating evidence suggests that the synchronization of canonical fast sleep spindles (i.e., spindles defined in young adults with a frequency of ≈ 12.5–16 Hz and a centro-parietal predominance; *Cox et al., 2017*) precisely during the up state of SOs provides an essential mechanism for neural communication, for example, supporting systems memory consolidation during sleep (*Hahn et al., 2020*; *Helfrich et al., 2018*; *Latchoumane et al., 2017*; *Mölle et al., 2002*; *Muehlroth et al., 2019*). Importantly, canonical fast spindles in turn assort hippocampal ripples (*Clemens et al., 2007*; *Helfrich et al., 2019*; *Siapas and Wilson, 1998*; *Staresina et al., 2015*), that code for wake experiences and are considered a reliable marker of hippocampal memory consolidation (*Buzsáki, 2015*; *Maingret et al., 2016*; *Sirota et al., 2003*). Moreover, canonical fast sleep spindles themselves are associated with facilitated hippocampal-neocortical connectivity (*Andrade et al., 2011*; *Cowan et al., 2020*). In addition to canonical fast sleep spindles, there is substantial evidence for a canonical slow sleep spindle type (i.e., spindles defined in young adults with a frequency of ≈ 9–12.5 Hz and a frontal predominance; *Cox et al., 2017*) in the human surface electroencephalogram (EEG; *De Gennaro and Ferrara, 2003*; *Fernandez and Lüthi, 2020*). That said, previous findings hint at a differential SO-slow spindle coupling pattern and their function is still elusive (*Klinzing et al.,*

*2016*; *Mölle et al., 2011*; *Muehlroth et al., 2019*; *Rasch and Born, 2013*). Taken together, the complex wave sequence of SO up state and canonical fast sleep spindles, together with hippocampal activity, is considered to provide the scaffold for the precisely timed reactivation of initially fragile hippocampal memory representations and their strengthening in neocortical networks (*Diekelmann and Born, 2010*; *Helfrich et al., 2019*; *Maingret et al., 2016*; *Staresina et al., 2015*). However, the precise coupling of sleep spindle activity to SOs, described above, does not seem to be fully present and functional from early childhood on. Recent evidence in rodents and humans indicates that the temporal co-ordination of SOs and spindles improves across childhood and adolescence (*García-Pérez et al., 2022*; *Hahn et al., 2020*; *Joechner et al., 2021*).

Likewise, the individual neural rhythms that define the coupling undergo substantial changes during child and adolescent development. Across maturation, sleep spindles increase in occurrence and their average frequency (*Purcell et al., 2017*; *Zhang et al., 2021*). Consistent with this, canonical slow sleep spindles were reported to mature and dominate during early childhood. In contrast, canonical fast spindles are rarely detected in young children and become increasingly present and clearly dissociable only around puberty (*D'Atri et al., 2018*; *Goldstone et al., 2019*; *Hoedlmoser et al., 2014*; *Shinomiya et al., 1999*). However, amongst others, the application of individually adjusted frequency bands revealed that already young children express functional fast spindles in centro-parietal sites, whereby these manifest in a development-specific fashion (*D'Atri et al., 2018*; *Friedrich et al., 2019*; *Joechner et al., 2021*; *Zhang et al., 2021*). Individualized rhythm detection methods provide an effective approach to capture true, dominant oscillatory rhythms despite substantial inter-individual variability, which presents a particular methodological challenge in developmental and age-comparative research (*Cox et al., 2017*; *Muehlroth and Werkle-Bergner, 2020*). While canonical fast spindles become more pronounced across development, an opposite trend can be observed for SOs (*Buchmann et al., 2011*; *Kurth et al., 2010a*). Slow neuronal activity is initially maximally expressed and originates over posterior areas, developing towards a mature anterior predominance (*Kurth et al., 2010b*; *Timofeev et al., 2020*). In summary, paralleling developmental changes in SO-spindle coupling, fast sleep spindles and SOs separately are manifested differentially across development. Nevertheless, it is still unclear how developments in sleep spindles and SOs interact to promote precise, adult-like temporal synchronization of sleep spindles during SOs across childhood and adolescence.

Therefore, we aimed to (i) characterize the modulation of sleep spindles during SOs across different ages and (ii) investigate how sleep spindle and SO maturity relate to the manifestation of SO-spindle coupling across different ages. Specifically, based on previous analyses (*Joechner et al., 2021*), we reasoned that the development of fast sleep spindles might be associated with the maturation of SO-spindle coupling. For this, we re-analyzed previously published nocturnal EEG data from a cross-sectional sample of 24 5- to 6-year-old children (13 female, $M_{age}$ = 5 years, 10.71 months, $SD_{age}$ = 7.28 months; *Joechner et al., 2021*) and a longitudinal cohort of 33 children tested at 8–11 years of age (T1; *Hoedlmoser et al., 2014*) and again at 14–18 years of age (T2; 23 female, $M_{ageT1}$ = 9 years, 11.70 months, $SD_{ageT1}$ = 8.35 months; $M_{ageT2}$ = 16 years, 4.91 months, $SD_{ageT2}$ = 9.06 months; *Hahn et al., 2019*; *Hahn et al., 2020*). Further, we examined a cross-sectional sample of 18 adults aged 20–26 years (15 female, $M_{age}$ = 21 years, 8.78 months, $SD_{age}$ = 20.04 months) as a reference sample for adult-like patterns. All cohorts underwent two nights of ambulatory polysomnography (PSG). Given well-known first night effects in children (*Scholle et al., 2003*), sleep spindles, SOs, and their coupling were only analyzed during the second night here.

## Results

### Dominant fast centro-parietal sleep spindles become more prevalent and increasingly resemble canonical fast sleep spindles with older age

In the first step, we aimed at identifying age-specific patterns of sleep spindles. To obtain evidence for two distinct dominant oscillatory spindle rhythms within all four age groups (5- to 6-, 8- to 11-, 14- to 18-, 20- to 26-year-olds), we firstly determined individual spindle peak frequencies (between 9–16 Hz) of background-corrected power spectra during non-rapid eye movement sleep (NREM, N2 and N3) at averaged frontal (F3, F4) and centro-parietal (C3, C4, Cz, Pz) electrodes, where slow and fast sleep spindles typically dominate, respectively (*Cox et al., 2017*; for background-corrected

power spectra, see *Figure 1A*; see *Supplementary file 1a and b* for statistical comparisons of peak frequencies). Based on the individual frontal and centro-parietal spindle peak frequencies, we then detected sleep spindle events in frontal and centro-parietal sites which represent the person- and age-specific dominant spindle rhythms per individual (see *Figure 1—figure supplement 1* for examples of averaged EEG signals time-locked to the occurrence of these dominant sleep spindles). We then compared features of these individually identified sleep spindles (i.e., frequency, density, amplitude; *Figure 1B–D*) between age groups (5- to 6-, 8- to 11-, 14- to 18-, 20- to 26-year-olds) and topographies (frontal, centro-parietal) using linear mixed-effects models (LMM):

$$\text{spindle feature} \sim 1 + \text{age group} * \text{topography} + (1|\text{ID})$$

The results from the LMMs were further specified using pairwise comparisons for which p-values were corrected using the Bonferroni method ($p_{adj}$; *Bland and Altman, 1995*).

For all features of individually identified, dominant sleep spindles, all main effects and the interaction between age group and sleep spindle topography reached significance ($F_{frequency}(3, 134.73) = 4.04$, $p_{frequency} = 0.009$; $F_{density}(3, 136.76) = 11.73$, $p_{density} < 0.001$; $F_{amplitude}(3, 140.21) = 5.45$, $p_{amplitude} = 0.001$; *Supplementary file 1c, e, and g*). For sleep spindle frequency, follow-up pairwise comparisons revealed that sleep spindles showed a higher frequency at centro-parietal derivations as compared to frontal electrodes within every age group (all $t \leq -5.38$, all $p_{adj} < 0.001$). Further, frontal spindle frequency was lowest for the 5- to 6- year-olds as compared to all older age groups (all $t \leq -3.69$, all $p_{adj} \leq 0.010$) and lower for the 8- to 11-year-olds as compared to the older age groups (all $t \leq -3.61$, all $p_{adj} \leq 0.013$), while there was no difference between the 14- to 18- and the 20- to 26-year-olds ($t(112.39) = 0.09$, $p_{adj} = 1.000$). For centro-parietal sleep spindles there was no frequency difference between the 5- to 6- and the 8- to 11-year-olds ($t(112.39) = -3.03$, $p_{adj} = 0.084$) and the 14- to 18- and the 20- to 26-year-olds ($t(112.39) = -2. 03$, $p_{adj} = 1.000$; for all pairwise comparison results, see *Supplementary file 1d*).

Hence, across all age groups, individually identified frontal spindles revealed lower frequencies than centro-parietal sleep spindles, indicating the presence of two distinguishably fast spindle types (note, the same applied to the peak frequencies, see *Supplementary file 1a and b*). Thus, we will henceforth be referring to slow, that is, frontal, and fast, that is, centro-parietal, spindles, respectively. Crucially, despite being faster, the frequency of the fast centro-parietal spindles was specific to the age of the participants in a way that at younger ages, these dominant fast sleep spindles were not yet in the range of canonical fast spindles (i.e., as on average observed in adults with a frequency of ≈ 12.5–16 Hz; *Cox et al., 2017*; see *Figure 1A–B*). For the vast majority of 14- to 18- and 20- to 26-year-olds, individually identified, dominant centro-parietal sleep spindles indeed matched the canonical fast sleep spindle frequency range (≈ 12.5–16 Hz, see *Figure 1A–B*). For the majority of children though, dominant fast centro-parietal spindles were nested in the canonical slow spindle range (i.e., as on average observed in adults with a frequency of ≈ 9–12.5 Hz; *Cox et al., 2017*) and thus manifested in a development-specific fashion. Hence, the term 'development-specific' will be employed to refer to individually determined, dominant fast centro-parietal sleep spindles across all age groups in the following. In contrast, since we also found dominant fast centro-parietal sleep spindles between 12.5 and 16 Hz in our adult sample, we will denote sleep spindle activity in this canonical fast spindle range and events detected exclusively within this canonical frequency range in centro-parietal sites as 'adult-like' in our sample across all age groups.

For density (*Figure 1C*; *Supplementary file 1f*), pairwise comparisons indicated higher density for slow frontal compared to development-specific fast centro-parietal spindles for the 5- to 6- ($t(136.76) = 5.69$, $p_{adj} < 0.001$) and the 8- to 11-year-olds ($t(136.76) = 4.62$, $p_{adj} < 0.001$), while the difference was not significant for 14- to 18- and the 20- to 26-year-olds ($t_{14-18}(136.76) = 0.83$, $p_{adj\ 14-18} = 1.000$; $t_{20-26}(136.76) = -2.07$, $p_{adj\ 20-26} = 1.000$). For spindle amplitude (*Figure 1D*, *Supplementary file 1h*), amplitudes were significantly higher for slow frontal as compared to development-specific fast centro-parietal spindles within all age groups (all $t \geq 4.74$, all $p_{adj} \leq 0.001$). The interaction was mainly driven by a significantly higher slow frontal amplitude for the 8- to 11-year-olds compared to all other age groups (all $-3.68 \geq t \geq 7.97$, all $p_{adj} \leq 0.009$), a higher slow frontal amplitude for the 5- to 6-year-olds compared to the 20- to 26-year-olds ($t(137) = 4.32$, $p_{adj} < 0.001$), a lower development-specific fast centro-parietal amplitude for the 14- to 18- compared to the 8- to 11-year-olds ($t(140) = 5.24$, $p_{adj} <$

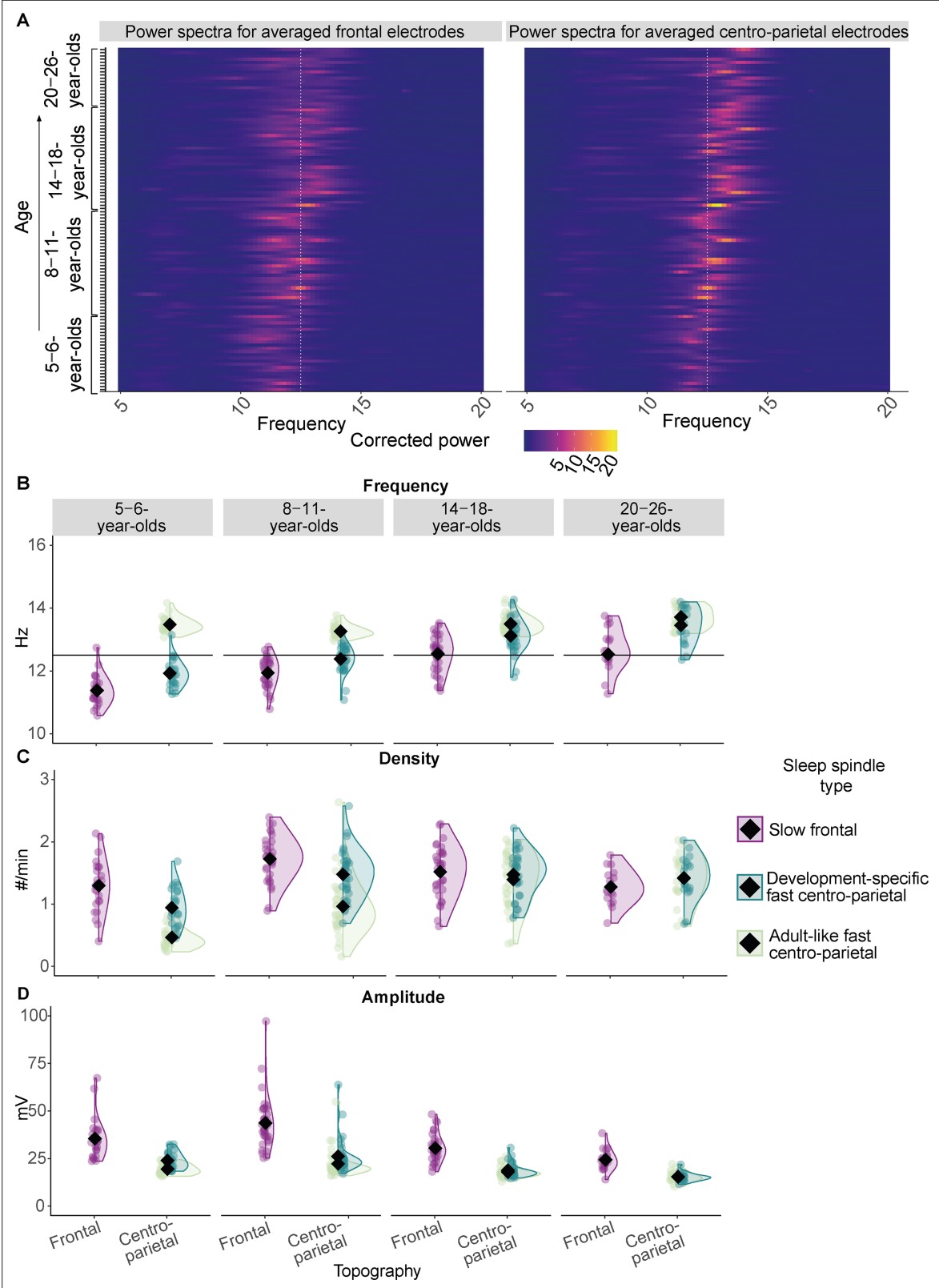

**Figure 1.** Sleep spindle features across the different age groups. (**A**) Background-corrected power spectra in averaged frontal (left) and averaged centro-parietal (right) electrodes for every participant at every test time point (y-axis). Power is coded by color for frequencies between 5–20 Hz (x-axis). The individual peak frequency in the sleep spindle frequency range (x-axis, 9–16 Hz) is reflected by brighter colors for every participant at every age (y-axis). Data are ordered by age from bottom to top (y-axis). The white dotted line at 12.5 Hz illustrates the frequency border for canonical fast sleep

*Figure 1 continued on next page*

*Figure 1 continued*

spindles (i.e., spindles defined in young adults with a frequency ≈ 12.5–16 Hz and a centro-parietal predominance; *Cox et al., 2017*). Note: Participants at an older age (top) showed higher peak frequencies. Further, a linear mixed-effects model revealed that peak frequencies in centro-parietal electrodes were significantly faster compared to frontal ones averaged across all age groups (*Supplementary file 1b*). However, for most of the young children (bottom part of the plot) the peak frequencies and the frequencies of the derived dominant fast spindles (**B**) were slower than the canonical fast sleep spindle band (i.e., development-specific). (**B–D**) Results from the comparison of adult-like fast (12.5–16 Hz) and individually identified slow frontal and development-specific fast centro-parietal sleep spindle (**B**) frequency, (**C**) density, and (**D**) amplitude for all four age groups. (**B**) Horizontal lines at 12.5 Hz illustrate the frequency border for canonical fast sleep spindles (i.e., spindles defined in young adults with a frequency ≈ 12.5–16 Hz and a centro-parietal predominance; *Cox et al., 2017*). Values are individual raw scores. Diamonds reflect estimated marginal means of the respective linear mixed-effects model. Statistical results can be found in *Supplementary files 1a–n*.

The online version of this article includes the following figure supplement(s) for figure 1:

**Figure supplement 1.** Exemplary averaged electroencephalography (EEG) signals time-locked to slow frontal (electrode F3), development-specific fast (electrode Cz), and adult-like fast (electrode Cz) sleep spindle (SP) centers for one 6-year-old participant as example for the 5- to 6-year-old age group, one participant who underwent EEG at 10 years of age and again at 16 years of age, as example for the 8- to 11- and 14- to 18-year-old age group, and one 21-year-old, as example for the 20- to 26-year-old age group.

**Figure supplement 2.** Example 30 second raw electroencephalography (EEG) window from a 6-year-old child with events that were detected as development-specific fast (solid circles) and adult-like fast (dashed circles) sleep spindles.

0.001), and a lower amplitude of the 20- to 26-year-olds compared to the 5- to 6- and 8- to 11-year-olds (t ≥ 3.34, $p_{adj}$ ≤ 0.030).

To sum up, our data indicate the pronounced existence of slow frontal and development-specific fast centro-parietal spindles at all ages under study. In particular, development-specific fast centro-parietal spindles were more numerous and faster at older age, especially compared to slow frontal sleep spindles. Given evidence for the specific role of canonical fast spindles for memory (*Rasch and Born, 2013*), henceforth, we focus on fast spindles detected at centro-parietal electrodes (however, corresponding analyses were also conducted for slow frontal spindles and can be found in *Supplementary file 1*).

After having characterized prevailing, development-specific fast centro-parietal spindles across all our age groups, we were interested in how they differed from canonical fast spindles, that is, those commonly, and also here, found in young adults with a frequency of ≈ 12.5–16 Hz (see *Figure 1* for the present adult sample, *Cox et al., 2017*; *Ujma et al., 2015*). Therefore, we additionally extracted fast spindles in centro-parietal electrodes by applying fixed frequency criteria between 12.5 and 16 Hz (henceforth, adult-like fast sleep spindles; see *Figure 1—figure supplement 2* for an example raw EEG trace with development-specific and adult-like fast spindles and *Figure 1—figure supplement 1* for examples of averaged EEG signals time-locked to development-specific and adult-like fast sleep spindles). Such adult-like fast spindles were shown before to be coupled to SOs in pre-school children, despite lacking evidence for their strong presence (*Joechner et al., 2021*; see also *Figure 1A*, Figure 3, and *Figure 1—figure supplement 1*). We then compared characteristics (i.e., frequency, density, amplitude; *Figure 1B–D*) of development-specific fast centro-parietal sleep spindles with adult-like fast centro-parietal spindles using LMMs and follow-up pairwise comparisons across the four age groups. In addition to 'age group', the factor 'spindle type' (development-specific, adult-like) was entered as a fixed factor:

$$\text{spindle feature} \sim 1 + \text{age group} * \text{spindle type} + (1|\text{ID})$$

Results indicated that all main effects and the interaction between the factors 'age group' and 'spindle type' were significant for frequency, density, and amplitude (for a summary of all analyses and all post-hoc comparisons see *Supplementary file 1i–n*). Post-hoc pairwise comparisons revealed that the frequency of adult-like fast sleep spindles was consistently higher as compared to the development-specific fast centro-parietal spindles within all age groups, except the 20- to 26-year-olds ($t_{5-6}$(135.85) = 18.70, $p_{adj\ 5-6}$ < 0.001; $t_{8-11}$(135.85) = 12.48, $p_{adj\ 8-11}$ < 0.001; $t_{14-18}$(135.85) = 5.37, $p_{adj\ 14-18}$ < 0.001; $t_{20-26}$(135.85) = 2.73, $p_{adj\ 20-26}$ = 0.199). In line with a generally higher frequency, in 5- to 6- and 8- to 11-year-olds, adult-like fast spindles had a lower amplitude compared to development-specific fast centro-parietal spindles ($t_{5-6}$(142.56) = –4.34, $p_{adj\ 5-6}$ < 0.001; $t_{8-11}$(142.56) = –4.47, $p_{adj\ 8-11}$ < 0.001). Crucially, this effect was non-significant in the two oldest age groups ($t_{14-18}$(142.56) = –0.99, $p_{adj\ 14-18}$ = 1.000; $t_{20-26}$(142.56) = 0.14, $p_{adj\ 20-26}$ = 1.000). Similarly, whereas the density of development-specific

fast sleep spindles was higher compared to adult-like fast spindle density in the 5- to 6- and the 8- to 11-year-olds ($t_{5\text{-}6}$(138.60) = –7.66, $p_{adj\ 5\text{-}6}$ < 0.001; $t_{8\text{-}11}$(138.60) = –9.69, $p_{adj\ 8\text{-}11}$ < 0.001), there was no difference in density for the 14- to 18- and 20- to 26-year-olds ($t_{14\text{-}18}$(138.60) = –1.50, $p_{adj\ 14\text{-}18}$ = 1.000; $t_{20\text{-}26}$(138.60) = –0.13, $p_{adj\ 20\text{-}26}$ = 1.000). Further, density of adult-like fast sleep spindles was higher for the two oldest age groups compared to the 8- to 11- and 5- to 6-year-olds and higher for the 8- to 11- as compared to the 5- to 6-year-olds (all t ≤ –4.15, all $p_{adj}$ ≤ 0.002). Density for the development-specific fast centro-parietal spindles was only lower for the 5- to 6-year-olds as compared to all older age groups ($t_{8\text{-}11}$(113.55) = –5.35, $p_{adj\ 8\text{-}11}$< 0.001; $t_{14\text{-}18}$(113.55) = –5.32, $p_{adj\ 14\text{-}18}$< 0.001; $t_{20\text{-}26}$(113.55) = –4.15, $p_{adj\ 20\text{-}26}$ = 0.002), while there was no difference between the 8- to 11-, the 14- to 18, and the 20- to 26-year-olds (0 < t ≤ 0.48; all $p_{adj}$ = 1.000; see *Supplementary file 1j, l, and n* for all comparisons).

Crucially, this indicates that despite the absence of a prominent peak in the power spectrum (*Figure 1A*), already young children express adult-like fast spindles which occur increasingly often with older age. Despite a higher frequency of the adult-like fast sleep spindles in the child and adolescent age groups, with older age, the differences in amplitude and density between development-specific and adult-like fast centro-parietal sleep spindles decreased and were no more evident in the adolescent and adult age groups. Hence, fast centro-parietal spindles become more numerous and adult-like in their frequency and amplitude characteristics in the course of maturation.

## Slow oscillations dominate anteriorly in older children, adolescents, and young adults but not in young children

Similar to the analyses on sleep spindles, we next compared features that define SOs (i.e., frequency, density, amplitude; *Figure 2*) between age groups and between SOs detected at frontal (averaged over F3, F4), centro-parietal (averaged over Cz, Pz), and occipital (cross-sectional child sample: Oz, longitudinal and cross-sectional adult samples: averaged over O1, O2) electrodes (topography factor) using LMMs and follow-up pairwise comparisons:

$$\text{SO feature} \sim 1 \ + \ \text{age group} \ * \ \text{topography} + (1|\text{ID})$$

Occipital locations were added based on observations suggesting a posterior dominance of SOs in young children (*Kurth et al., 2010b*; *Timofeev et al., 2020*). Further, considering evidence indicating that surface SOs might be most powerful medially (*Murphy et al., 2009*), we focused on midline electrodes, while also keeping the overlap of electrodes between our samples as high as possible.

Results revealed that the frequency of SOs differed over topographic locations across age groups (age group*topography interaction: F(6,238.15) = 13.10, p < 0.001; see also *Supplementary file 1o*). Pairwise comparisons (*Supplementary file 1p*) indicated that SO frequency differed between recording sites for the 14- to 18- and the 20- to 26-year-olds. For both age groups frontal and centro-parietal frequency was higher compared to occipital frequency ($t_{\text{frontal-occipital }14\text{-}18}$(238.15) = 10.48, $p_{adj\ \text{frontal-occipital }14\text{-}18}$ < 0.001; $t_{\text{centro-parietal-occipital }14\text{-}18}$(238.15) = 8.09, $p_{adj\ \text{centro-parietal-occipital }14\text{-}18}$ < 0.001; $t_{\text{frontal-occipital }20\text{-}26}$(238.15) = 9.62, $p_{adj\ \text{frontal-occipital }20\text{-}26}$ < 0.001; $t_{\text{centro-parietal-occipital }20\text{-}26}$(238.15) = 7.26, $p_{adj\ \text{centro-parietal-occipital }20\text{-}26}$ < 0.001; *Figure 2A*). For SO density, summary statistics indicated significant main effects for 'age group' (*Supplementary file 1q*; F(3,96.72) = 64.23, p < 0.001) and 'topography' (F(2,240.58) = 4.29, p = 0.015) but no interaction effect. Pairwise comparisons revealed that density, averaged across topographical recording sites, was lower for the 8- to 11- compared to both the 5- to 6- (t(78.34) = 3.30, $p_{adj}$ = 0.009) and the 14- to 18-year-olds (t(240.58) = –13.64, $p_{adj}$ < 0.001) and lower for the 20- to 26-year-olds compared to the 14- to 18-year-olds (t(78.34) = 4.27, $p_{adj}$ < 0.001; see *Supplementary file 1r* for all pairwise comparisons). Further, density averaged across all age groups was significantly higher for frontal as compared to centro-parietal derivations (t(241) = 2.93, $p_{adj}$ = 0.011; *Figure 2B*). For SO amplitude, inspection of the summary statistics and the pairwise comparisons revealed that all age groups, except the 5- to 6-year-olds, expressed SOs at a higher amplitude at frontal as compared to both centro-parietal and occipital locations (all t ≥ 6.68, all p < 0.001; see *Supplementary file 1s and t* for all results; *Figure 2C*).

To summarize, we did not observe a frontal prevalence of SOs in the youngest children. However, this dominance was present in all older age groups. Therefore, similar to our sleep spindle analyses, we concentrate on centro-parietal SOs for all subsequent analyses. Results for spindles and SOs in additional topographical locations can be found in *Supplementary file 1* and the figure supplements.

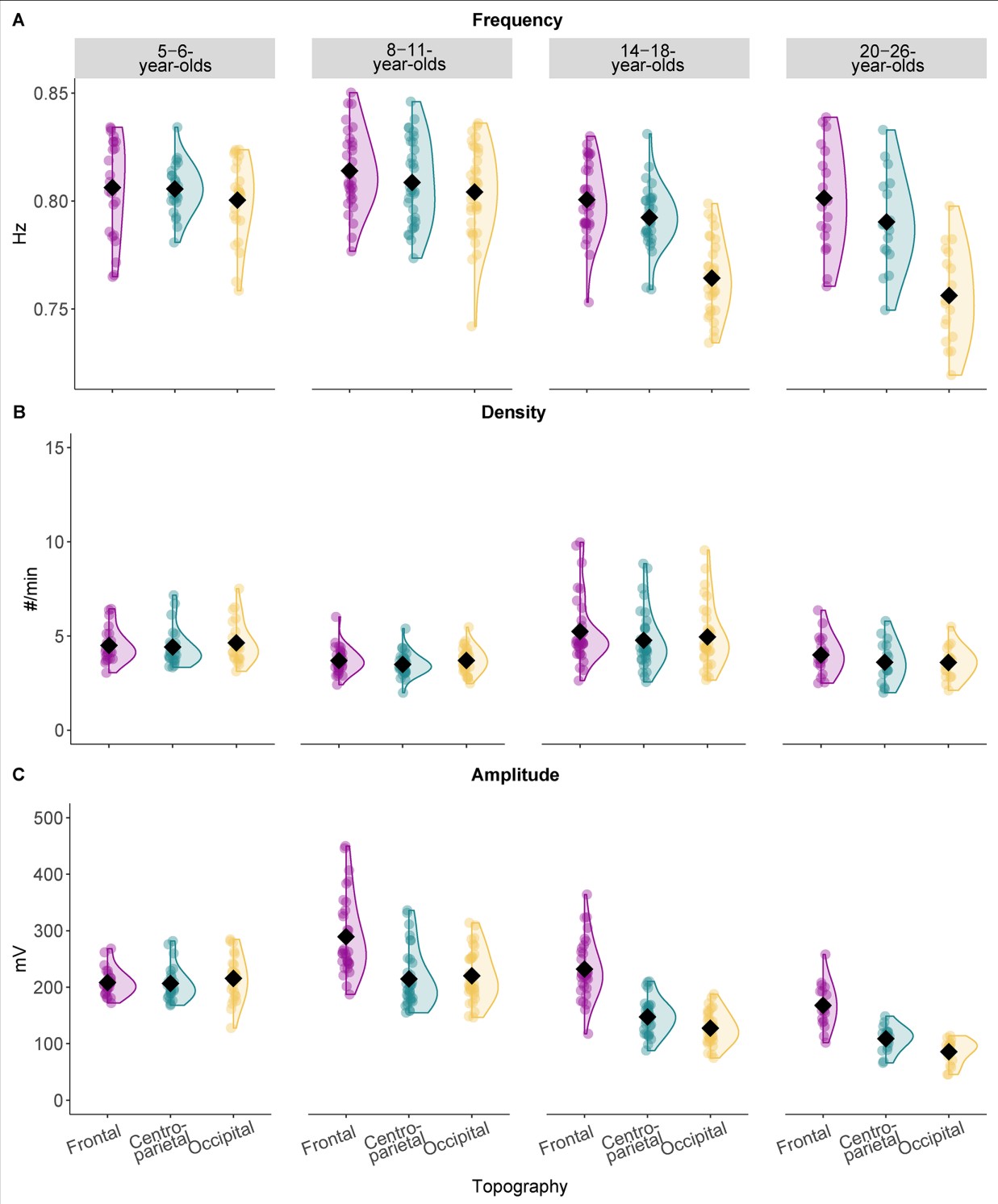

**Figure 2.** Slow oscillation features across the different age groups. Results from the comparison of slow oscillation (**A**) frequency, (**B**) density, and (**C**) amplitude for all four age groups. Values are individual raw scores. Diamonds reflect estimated marginal means of the respective linear mixed-effects model. Note that we observed the strongest age-related differences for the slow oscillation amplitude (**C**) in different topographic locations. Statistical results can be found in ***Supplementary file 1o–t***.

**Figure 3.** Centro-parietal power differences between trials with and without centro-parietal slow oscillations (in t-score units). Significant clusters are outlined in black (cluster-based permutation test, cluster α < 0.05, two-sided test). Warmer colors indicate higher power during trials with slow oscillations and colder colors indicate lower power during trials with slow oscillations as compared to trials without slow oscillations. The average centro-parietal slow oscillation for each age group (**A–D**) is plotted onto the power differences in black to illustrate the relation to slow oscillation phase (scale in mV on the right y-axis of each plot). The sleep spindle frequency range is highlighted by the dashed window. Note that the strongest power increases during slow oscillations were observed in a frequency range reflecting the adult-like fast sleep spindle range (12.5–16 Hz). Results for frontal power and frontal slow oscillations can be found in *Figure 3—figure supplement 1*.

The online version of this article includes the following figure supplement(s) for figure 3:

**Figure supplement 1.** Power differences between trials with and without slow oscillations (in t-score units) in different topographical locations.

## Temporal modulation of adult-like fast sleep spindle power during slow oscillations is present across all ages

We observed robust age-related differences in sleep spindles and SOs across the four age groups. But do they also affect the temporal coupling between these two neural rhythms at a given age? In the first step, we aimed at determining the spectral and temporal characteristics of SO-coupled spindles. Hence, within the four age groups, we compared spectral power (5–20 Hz) over centro-parietal recording locations during trials with and without centro-parietal SOs (±1.2 s around SO down peak and equally long trials without SOs) using cluster-based permutation tests (*Maris and Oostenveld, 2007*). Within all age groups, power in a broad range including the sleep spindle frequency band (9–16 Hz) was significantly higher during SOs as compared to trials without SOs (all cluster p < 0.001), suggesting temporal clustering of spindles during the SO cycle (*Figure 3*). On a descriptive level, the strongest power differences during SOs as compared to trials without SOs for all four age groups were located within the frequency range of adult-like fast sleep spindles (12.5–16 Hz) within one second after the down peak, close to the up peak (*Figure 3*). However, the modulation of power in this adult-like fast spindle frequency range seemed to be stronger and more precise during the SO up peak in our two oldest age groups. Specifically, for the younger children, the increased power in the adult-like fast spindle frequency range was surprising, given the overall lower density and power of adult-like fast sleep spindles in 5- to 6- and also a majority of 8- to 11-year-old children (*Figure 1*).

## At older age, development-specific fast sleep spindle occurrence is more strongly modulated by slow oscillations while slow oscillation-adult-like fast spindle coupling becomes temporally more precise

Having identified evidence for the temporally ordered occurrence of sleep spindles during specific SO phases in all our age groups, in the next step, we were interested in the precise temporal co-ordination of spindle events and SOs. Therefore, we created peri-event time histograms (PETH) of development-specific fast spindles during centro-parietal SOs (± 1.2 s around the SO down peak), showing the occurrence of sleep spindles within 100 ms time bins during the SO down peak-centered time window. To identify patterns of increased and decreased spindle occurrence during the SO cycle, we compared the resulting sleep spindle occurrence-percentage distribution per participant with a participant-specific surrogate distribution using cluster-based permutation tests (*Maris and Oostenveld, 2007*).

As can be inferred from *Figure 4A*, we observed a shift towards a clear coupling from the 5- to 6- to the 14- to 18-year-olds. Only the 14- to 18-year-olds reliably presented the coupling pattern known from, and also observed here in, adults (see *Figure 4A*, rightmost plot): A decreased sleep spindle probability during the SO down state and an increased occurrence during the SO up state.

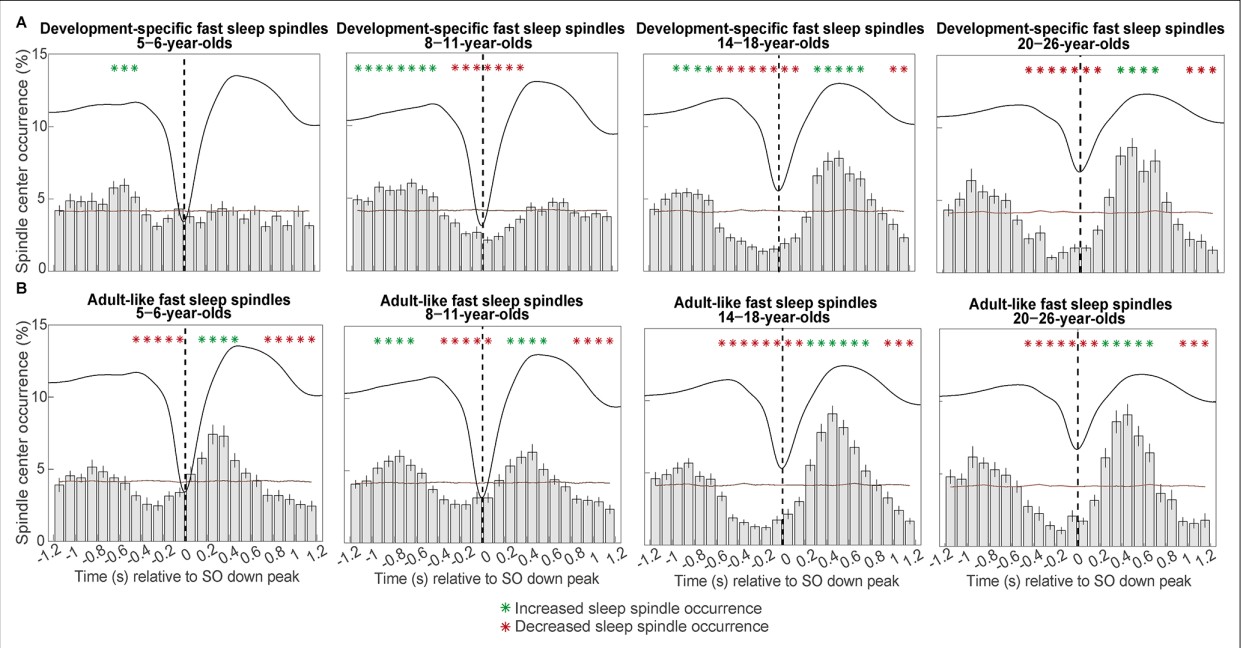

**Figure 4.** Peri-event time histograms for (**A**) development-specific fast centro-parietal spindles and (**B**) adult-like fast centro-parietal spindles showing the proportion of events occurring within 100 ms bins during centro-parietal slow oscillations. Error bars represent standard errors. Green asterisks mark increased spindle occurrence (positive cluster, cluster-based permutation test, cluster α < 0.05, two-sided test) and red asterisks mark decreased spindle occurrence (negative cluster, cluster-based permutation test, cluster α < 0.05, two-sided test) compared to a surrogate distribution representing random occurrence (horizontal line). The dashed vertical line indicates the slow oscillation down peak. The average centro-parietal slow oscillation of each age group is shown in black to illustrate the relation to the slow oscillation phase. Results for slow oscillations and spindles in different topographies can be found in *Figure 4—figure supplement 1*.

The online version of this article includes the following figure supplement(s) for figure 4:

**Figure supplement 1.** Peri-event time histograms for (**A–B**) slow frontal, (**C**) development-specific fast centro-parietal, and (**D**) adult-like fast spindles showing the proportion of sleep spindles occurring within 100 ms bins during (**A**) centro-parietal and (**B–D**) frontal slow oscillations.

**Figure supplement 2.** Comparison of the peri-event time histogram results based on (**A**) all available adult-like fast sleep spindles and (**B**) when run with a randomly resampled subset of half the number of adult-like fast sleep spindles (100 times randomly resampled and averaged across the 100 resamples).

**Figure supplement 3.** Co-occurrence data of slow frontal and development-specific fast centro-parietal sleep spindles and slow oscillations in different topographical locations.

**Figure supplement 4.** Co-occurrence data of adult-like fast sleep spindles and slow oscillations in different topographical locations.

Statistically, we found one positive cluster (p = 0.001) for the 5- to 6-year-olds from −700 to −400 ms for the modulation of development-specific fast centro-parietal spindles (SO up peak at ≈ 465 ms). For the 8- to 11-year-olds, we identified a more extended positive cluster from −1200 to −400 (p < 0.001) and one negative cluster from −300 to 400 ms (p < 0.001) for development-specific fast centro-parietal spindles (SO up peak at ≈ 488 ms). For the 14- to 18-year-olds, the analyses indicated two positive clusters. One from 300 to 800 ms (p = 0.002) and another from −1000 to −600 ms (p = 0.014) and two negative clusters from −600 to 200 ms (p < 0.001) and from 1000 to 1200 ms (p = 0.034) for development-specific fast centro-parietal spindles (SO up peak at ≈ 532 ms). For the 20- to 26-year-olds, there was one significant positive cluster (p = 0.005) from 300 to 700 ms and two negative clusters; one from −500 to 200 ms (p < 0.001) and from 900 to 1200 ms (p = 0.004; SO up peak at ≈ 559 ms).

Critically, and somewhat unexpectedly for the two younger age groups, the previous analysis of spectral power differences in the sleep spindle frequency range along the SO cycle suggested temporal SO-spindle alignment specifically for events in the adult-like fast spindle range (*Joechner et al., 2021*; *Piantoni et al., 2013a*). Hence, we repeated the PETH analyses for adult-like fast sleep spindles. As can be inferred from *Figure 4B*, we observed a clear coupling of adult-like fast spindle

occurrence rates to specific phases of the SO cycle within all age groups— although temporally less precisely during the SO down and up peaks in the younger age groups.

For 5- to 6-year-olds, we identified one positive cluster (p < 0.001) from 100 to 500 ms and two negative clusters (both p < 0.001) from –500 to 0 ms and from 700 to 1200 ms for adult-like fast centro-parietal spindles (SO up peak at ≈ 465 ms). For the 8- to 11-year-olds, we found two positive clusters from –1000 to –600 ms (p < 0.001) and from 200 to 600 ms (p = 0.001) and two negative clusters (both p < 0.001) from –400 to 100 ms and from 800 to 1200 ms (SO up peak at ≈ 488 ms). For the 14- to 18-year-olds, we identified one positive cluster (p < 0.001) from 200 to 800 ms and two negative clusters from –600 to 200 ms (p < 0.001) and from 900 to 1200 ms (p = 0.004) for adult-like fast centro-parietal spindles (SO up peak at ≈ 532 ms). Lastly, for the 20- to 26-year-olds, analyses revealed one positive cluster (p = 0.003) from 200 to 700 ms and two negative clusters. One cluster (p < 0.001) from –500 to 200 ms and another one (p = 0.004) from 900 to 1200 ms (SO up peak at ≈ 559 ms). Hence, while occurrence patterns during SOs appeared different for development-specific and adult-like fast sleep spindles in the child age groups, modulation patterns during SOs were highly comparable for these two fast spindle types in the two older age groups.

To summarize, a clear coupling of spindle occurrence at specific phases of the SO could hardly be detected for development-specific fast centro-parietal sleep spindles in the younger age groups. Importantly, adult-like fast sleep spindles were already modulated by SOs in the younger age groups— even though they only occurred very rarely. Of note, a randomly selected lower absolute number of adult-like fast sleep spindles in the 14- to 18-year-olds did not affect the co-occurrence pattern, suggesting that enhanced modulation of sleep spindles during SOs does not merely depend on the number of sleep spindles (see *Figure 4—figure supplement 2*). However, independent of the sleep spindle type, increased sleep spindle occurrence was more precisely timed with the SO up peak at older ages. Hence, despite clearly identifiable development-specific fast spindles, pronounced and temporally precise SO-spindle coupling seems to depend on the presence of adult-like fast sleep spindles.

## Slow oscillation-spindle coupling is related to sleep spindle maturity

So far, in line with previous observations (*Joechner et al., 2021*), both analyses on SO-spindle co-occurrence suggested that sleep spindles in the adult-like fast frequency range rather than the more dominant, development-specific fast sleep spindles are coupled to SOs within all age groups. Hence, we reasoned, that the maturation of dominant, development-specific fast sleep spindles towards adult-like fast spindles may explain age-related differences in the strength and temporal precision of SO-spindle coupling patterns.

To capture the 'maturational stage' of fast spindles for every participant, we resorted to our observed age-related differences in sleep spindles (*Figure 1*) and computed the distance between development-specific and adult-like fast sleep spindles within each individual at a given age. Specifically, we calculated difference measures in sleep spindle characteristics (i.e., frequency, density, amplitude) between adult-like and development-specific fast sleep spindles. Note, given the opposing signs of the fast spindle maturity differences, we inverted the frequency difference values. Further, we re-scaled the frequency, density, and amplitude difference scores using Z-transformation across all participants to convert all metrics into a common space. As illustrated in *Figure 5A–C*, the increasing Z-transformed difference scores (henceforth termed 'spindle maturity scores') with older age of the participants capture the fact that the dominant, development-specific fast centro-parietal sleep spindles resemble more closely adult-like fast spindles at older ages.

Following a similar logic, we also created a measure for SO maturity. Based on our observation that age differences were mostly reflected in the emerging frontal dominance of SOs with older age (*Figure 2*) and on the literature (e.g., *Kurth et al., 2010b*; *Timofeev et al., 2020*), we calculated the difference between the amplitude in frontal and centro-parietal regions to reflect the maturity pattern of SOs within a participant. In parallel with the procedure for the fast spindle maturity scores, we Z-standardized the SO difference measure to be in the same metric space as the spindle maturity values for all subsequent analyses. Comparably to the fast spindle maturity scores, participants of older age showed higher Z-standardized SO difference scores (*Figure 5D*).

To examine how fast sleep spindle maturity would be associated with SO-spindle coupling patterns across different ages, we conducted two analyses. For one, we examined the relation between the

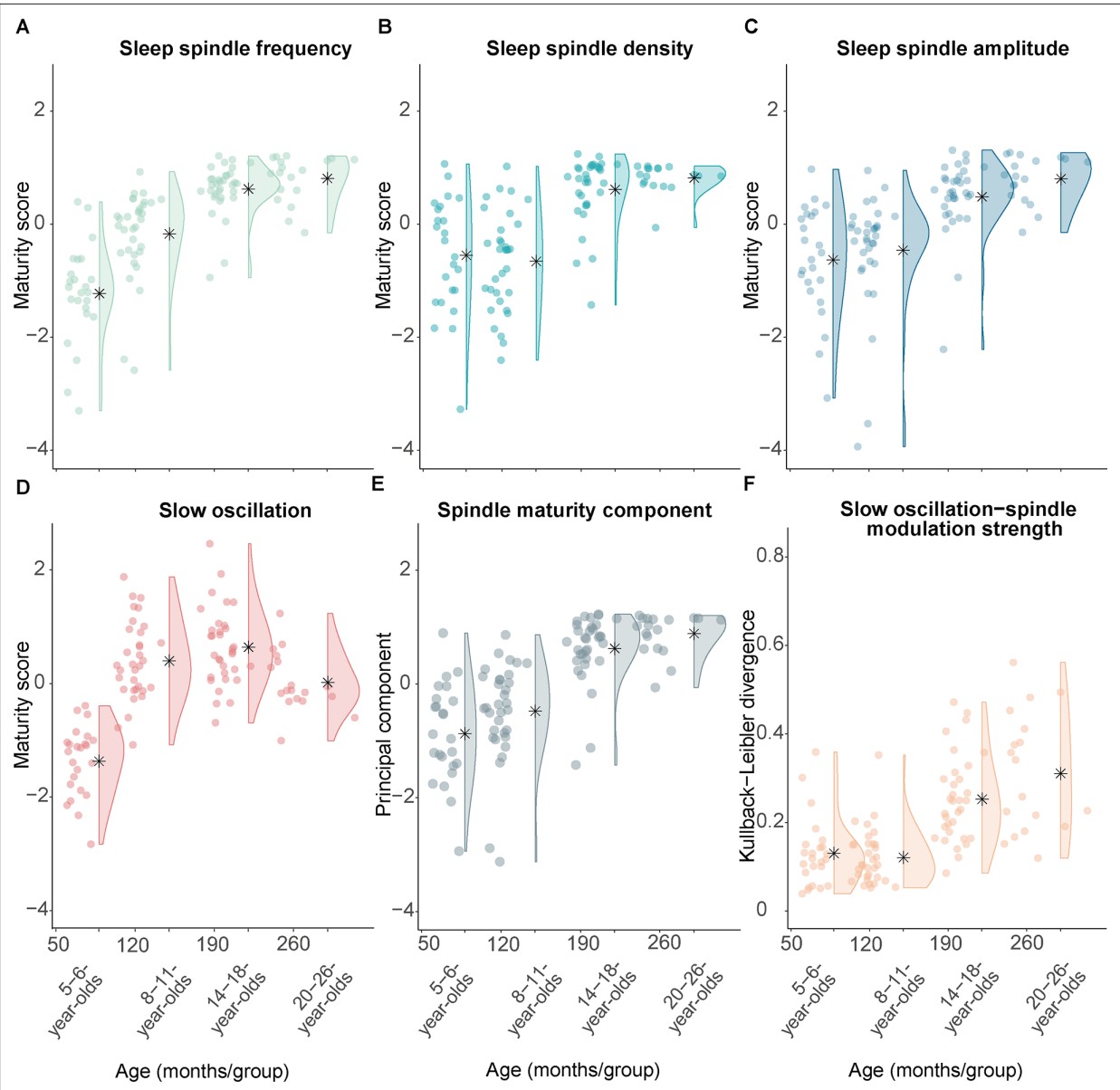

**Figure 5.** Measures of sleep spindle and slow oscillation maturity and slow oscillation-spindle coupling strength across all age groups. (**A–C**) Fast sleep spindle maturity scores for (**A**) frequency, (**B**) density, and (**C**) amplitude. Maturity scores reflect the Z-standardized differences between adult-like and development-specific fast centro-parietal sleep spindles. (**D**) Slow oscillation maturity scores represent the Z-standardized difference between frontal and centro-parietal amplitudes. (**E**) First principal component of a principal component analysis on the three fast spindle maturity scores which are shown in (**A–C**). (**F**) Kullback-Leibler divergence for development-specific fast centro-parietal sleep spindle modulation during centro-parietal slow oscillations, reflecting slow oscillation-spindle modulation strength. For all measures, higher values are linked to older age. Asterisks illustrate the mean.

pattern of power modulations in the spindle frequency range (9–16 Hz) during the complete SO trial (down peak ± 1.2 s, time-frequency t-maps, *Figure 3*) with the fast spindle maturity scores for sleep spindle frequency, density, and amplitude using a partial least squares correlation (PLSC; *Krishnan et al., 2011*) across all participants. This analysis provides pairs of SO-spindle coupling profiles and associated, multivariate patterns of spindle maturity scores (spindle maturity profile). Based on a permutation test, we identified one significant pair of a spindle maturity profile (*Figure 6A*) and a specific time-frequency SO-spindle coupling pattern (*Figure 6B*; singular vector pair p < 0.001). All fast spindle maturity measures contributed reliably and positively to this significant, positive correlation (as indicated by the direction of values in the spindle maturity profile and the non-zero crossing confidence intervals in *Figure 6A*). As can be inferred from *Figure 6B*, the SO-spindle coupling pattern

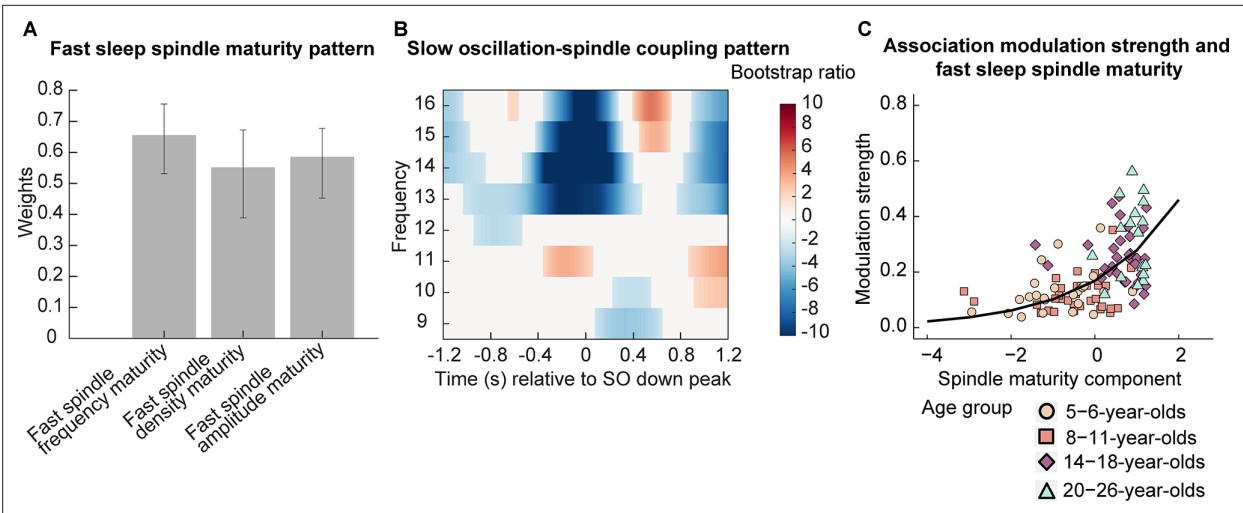

**Figure 6.** Association between fast spindle maturity and centro-parietal slow oscillation-spindle coupling. (**A**–**B**) Results from a partial least squares correlation revealed one (**A**) fast spindle maturity profile significantly associated with (**B**) a centro-parietal slow oscillation-spindle coupling pattern. (**A**) Weights of the first singular vector dimension of the fast spindle maturity scores. Error bars represent 95% bootstrap confidence intervals. (**B**) Weights of the first singular vector dimension of the slow oscillation-spindle coupling pattern by means of bootstrap ratios. Only values > 1.96 and < −1.96 are colored. Warmer colors represent higher power and colder colors reflect lower power. (**C**) Scatterplot of the association between the modulation strength (Kullback-Leibler divergence) of development-specific fast centro-parietal sleep spindles during centro-parietal SOs and the fast spindle maturity component. The curved line represents the prediction from the generalized linear mixed-effects model for the simple effect of the spindle maturity component. For visualization purposes, the four age groups are indicated by different shapes and colors. Results for frontal slow oscillations can be found in *Supplementary file 1u* and *Figure 6—figure supplement 1*.

The online version of this article includes the following figure supplement(s) for figure 6:

**Figure supplement 1.** Association between fast spindle maturity and frontal slow oscillation-development-specific fast spindle coupling.

suggests, that higher fast sleep spindle maturity in all features (*Figure 6A*) was associated with a more adult-like SO-spindle coupling pattern, reflected in: (i) lower adult-like fast spindle power and higher power in the canonical slow sleep spindle range during the down state and (ii) higher power in the adult-like fast spindle frequency range during the SO up peak. Overall this indicates that a stronger presence of spindles with more adult-like fast spindle characteristics is associated with the well-known pattern of increased adult-like fast sleep spindle activity during the SO up peak, increased canonical slow spindle activity during the down state, and decreased activity of adult-like fast spindles during the down peak.

After having identified that fast spindle maturity was associated with stronger modulation of spindle power during the SO cycle, we further examined whether fast spindle maturity would also be related to the strength of modulation of development-specific fast centro-parietal sleep spindles during SOs (see PETH analyses, *Figure 4A*). Since all three indicators of fast sleep spindle maturity (frequency, density, amplitude maturity) showed age differences and were related to the pattern of spindle power modulations during SOs (*Figure 6A*), we used principal component analysis (PCA) to create one latent component capturing the maximal amount of variance across our fast spindle maturity indicators (as for individual sleep spindle maturity scores, higher component values indicate higher fast spindle maturity, *Figure 5E*). This fast spindle maturity component was then used to examine the relation between fast spindle maturity and development-specific fast spindle modulation strength across all participants at every age. We quantified the strength of modulation of development-specific fast centro-parietal sleep spindles during centro-parietal SOs using the Kullback-Leibler (KL) divergence for which higher values reflect a stronger clustering of sleep spindles around specific phases of SOs (*Figure 5F*, higher values are associated with older age). We then conducted a log-linked gamma generalized LMM (GLMM) analysis with the KL divergence during centro-parietal SOs as the dependent variable and the fast spindle maturity component as a fixed effect, allowing for a random intercept for each participant. We further added the SO maturity score as a covariate. The GLMM revealed that only higher fast spindle maturity was associated with stronger modulation of development-specific fast centro-parietal sleep spindles during SOs ($\beta$ = 0.50, t = 8.97, p < 0.001; see

*Figure 6C*), while SO maturity was not significantly related to the modulation strength (β = 0.09, t = 1.28, p = 0.202). These results did also hold when controlling for age (*Supplementary file 1v*). In sum, our results suggest that developmental age differences in the manifestation of SO-spindle coupling are uniquely related to the degree to which the dominant, development-specific fast spindle type within an individual, that is, the sleep spindles that can be identified as peaks in the power spectrum (*Aru et al., 2015*; see *Figure 1A*), shares characteristics with adult-like fast sleep spindles.

## Discussion

Although the synchronization of sleep spindles by the up peak of SOs and its functional significance has been recognized for decades, its development remains elusive (*Muehlroth et al., 2019*; *Schreiner et al., 2021*; *Staresina et al., 2015*; *Steriade, 2006*). Based on within-person detection of single events, we provide a detailed characterization of age-specific patterns of slow and fast sleep spindles, SOs, and their coupling in children aged 5–6 and 8–11 years, in adolescents aged 14–18 years, and in young adults aged 20–26 years. Specifically, in the child age groups, we noted that the predominant type of fast sleep spindles, as characterized by peaks in the power spectrum, is found in a frequency range slower than known from research in adults (canonical range, 12.5–16 Hz; *Cox et al., 2017*) and found also here in the adult sample (i.e., they manifest development-specifically). Surprisingly, the inspection of SO-spindle coupling patterns suggested synchronization being driven by spindles in the adult-like canonical fast spindle range—even in the younger age groups but notably less precisely timed during the SO cycle. Additional single event detection restricted to the canonical fast spindle range indeed revealed adult-like fast centro-parietal sleep spindle events in all age groups, although with only minor presence in the two child cohorts. Interrogation of coupling precision by means of PETHs confirmed that the coupling pattern found in adults, that is, reduced spindle occurrence during the down peak and increased sleep spindle likelihood precisely during the up state, can only be found for adult-like fast spindles and is less pronounced for the prevailing, development-specific fast spindles found in the child age groups.

To further corroborate these observations, we determined personalized measures for fast sleep spindle maturity based on the differences between the predominant, development-specific and the adult-like fast sleep spindles in terms of frequency, amplitude, and density. Indeed, we observed that higher fast spindle maturity (i.e., higher spindle maturity scores) was associated with a stronger SO-spindle modulation pattern of (i) decreased adult-like fast spindle power during the down state, (ii) increased adult-like fast spindle power during the up state, and (iii) increased canonical slow spindle power during the down peak. Most importantly, more advanced fast spindle maturity was exclusively linked to stronger modulation of development-specific fast centro-parietal spindles during SOs, over and above SO maturity. Hence, the present results provide evidence that the development of temporally precise SO-spindle coupling may specifically be linked to the maturation of fast sleep spindles.

Overall, our findings suggest that the temporally precise synchronization of sleep spindles during the up peak of SOs might not be an inherent feature of the thalamocortical system. Rather, SO-spindle coupling patterns differ systematically across the lifespan (*Hahn et al., 2020*; *Helfrich et al., 2018*; *Joechner et al., 2021*; *Muehlroth et al., 2019*), likely depending on age-specific anatomical and electrophysiological properties of the thalamocortical network.

Previous research in older adults suggested that age-related dispersion and imprecision of frontal SO-spindle coupling were related to atrophy in the prefrontal cortex and the thalamus—brain sites critically involved in the generation of mature SOs and spindles, respectively (*Helfrich et al., 2018*; *Muehlroth et al., 2019*). While these studies indirectly hypothesized that age-related changes in fast sleep spindle and SO features may contribute to alterations in the SO-spindle coupling pattern, we provide direct evidence that the development of the well-known coupling of sleep spindles to SO up peaks is uniquely linked to the emergence of an increasing number of spindles in the canonical fast sleep spindle frequency range. However, the exact anatomical and functional developments underlying the increased emergence of canonical fast spindles and their modulation by SOs remain to be elucidated. Furthermore, the question arises whether development-specific and canonical fast sleep spindles represent distinct representations of the same though developing generating network or whether they originate through different structures and connections.

Sleep spindles arise within well-described thalamic nuclei and thalamocortical circuits (*Fernandez and Lüthi, 2020*; *Steriade, 1995*). For one, the duration of hyperpolarization and the resulting length

of hyperpolarization-rebound sequences in thalamocortical cells has been reported to particularly account for slower and faster spindle frequencies (*Steriade, 2003*; *Steriade and Llinás, 1988*). Hence, decreasing hyperpolarization in thalamocortical, and potentially reticular thalamic and cortical, cells may underly the global increase in the frequency of predominant sleep spindles across child and adolescent development (*Campbell and Feinberg, 2016*; *Zhang et al., 2021*). A relatively stronger age-related drop in hyperpolarization in topographically selected assemblies may lead to the expression of canonical fast spindles over posterior scalp electrodes. While the overall state of the brain determines the excitability of thalamocortical cells (*Steriade and Llinás, 1988*), it is still elusive which cellular or other developments could drive age-related alterations in thalamocortical hyperpolarizations.

Further, both thalamic nuclei and thalamocortical connectivity are refined across development (*Steiner et al., 2020*). This applies to structural and functional changes with studies indicating increased white matter and functional connectivity of thalamocortical tracts with older age—specifically with frontal, motor, and somatosensory cortical areas (*Avery et al., 2022*; *Fair et al., 2010*; *Steiner et al., 2020*). Previously, inter-individual differences in magnetic resonance imaging indicators of white matter properties, supposed to index myelin-dependent efficiency in signal transmission (*Chanraud et al., 2010*), were linked to the expression of spindle power and frequency (*Mander et al., 2017*; *Piantoni et al., 2013b*; *Sanchez et al., 2020*; *Vien et al., 2019*). Therefore, increased white matter integrity of certain thalamocortical projections may support the increase in sleep spindle frequency and higher numbers of fast sleep spindles with older age during development.

Importantly, accumulating findings support the conjecture that primarily canonical fast sleep spindles coalesce with the up peak of SOs, while slow sleep spindles are synchronized more towards the down peak (*Bastian et al., 2022*; *Kurz et al., 2021*; *Mölle et al., 2011*; *Muehlroth et al., 2019*). If merely the absolute frequency range of spindles would determine the coupling pattern, one might have simply expected to find a phase shift in coupling with older age and accompanying faster sleep spindles. However, in none of the age groups did we observe strong evidence indicating that sleep spindles outside of the canonical fast spindle frequency range or at other topographic locations couple to distinct phases of the SO. Rather than a synchronization of development-specific fast sleep spindles during the down state, we found a general pattern of lower spindle modulation in 5- to 6-year-olds where development-specific fast spindle frequency is the lowest. However, development-specific fast sleep spindles were not only lower in frequency but also occurred less often in children. Surprisingly though, the modulation of adult-like fast sleep spindles during SOs was well detectable across all age groups with this co-occurrence pattern becoming more precisely timed with SO up and down peaks with older age—despite huge age-related differences in their occurrence rate and the fact that adult-like fast sleep spindles showed very low density in both child cohorts (see *Figure 1*). Therefore, our data suggest that neither frequency nor density alone explains SO-spindle coupling patterns. Rather, in individuals increasingly capable of expressing adult-like, canonical fast sleep spindles, spindles at all frequencies are globally modulated more strongly and in a comparable fashion. However, it remains an open question how the distinct coalescence of canonical fast and slow spindles with SOs develops.

While sleep spindles can be solely initiated in intra-thalamic circuits and thus could co-occur with SOs merely by chance, corticothalamic input is one of the most potent mechanisms inducing spindles (*Bonjean et al., 2011*; *Contreras and Steriade, 1995*; *Helfrich et al., 2018*). The connectivity between the cortex and the thalamus is supposed to explain the formation of SO-spindle modulation in a way that the synchronous firing of cortical cells during the depolarizing SO up state can excite thalamic sleep spindle generation while the joint neuronal hyperpolarization during the down state terminates and inhibits spindles (*Contreras and Steriade, 1995*; *Steriade, 2006*). Hence, it is conceivable that enhanced SO-spindle coupling not only indexes the development of thalamocortical pathways (that may give rise to sleep spindles at faster frequencies) but also changes in corticothalamic connectivity allowing for increasingly efficient cortical control over spindle generation—specifically of canonical fast spindles (*Contreras and Steriade, 1995*).

While in our analyses SO maturity did not significantly explain SO-spindle coupling patterns over and above fast sleep spindle maturity, there is evidence indicating that SO characteristics are associated with the pattern of SO-spindle co-occurrence. For instance, a higher SO amplitude was associated with coupling strength (*Kurz et al., 2021*). Further, in older adults, a diminished directional influence of SOs was related to imprecise SO-spindle coupling (*Helfrich et al., 2018*). Therefore, we

cannot exclude a potentially important contribution of the development and characteristics of SOs for SO-spindle coupling. It may just be that the prominent age-related differences in fast sleep spindles across childhood and adolescence may be relatively more influential and/or that SO features other than the ones examined here may be effective in shaping SO-spindle coupling.

On a functional level, the precisely timed synchronization of sleep spindles during SO up peaks is in particular discussed as important for memory consolidation (*Maingret et al., 2016*; *Niknazar et al., 2015*). In line with this, developmental differences in SO-spindle coupling from late childhood to adolescence have been linked to developmental enhancements in memory consolidation (*García-Pérez et al., 2022*; *Hahn et al., 2020*), implying less functional consolidation with less pronounced SO-spindle coupling (*Helfrich et al., 2018*; *Muehlroth et al., 2019*). Hence, besides the development of thalamocortical and corticothalamic networks, SO-spindle coupling might also reflect the functional maturation of hippocampal-neocortical networks (*Cowan et al., 2020*; *Helfrich et al., 2018*; *Muehlroth et al., 2019*). Yet, already children can show extraordinary levels of sleep-associated memory consolidation (e.g., *Peiffer et al., 2020*; *Urbain et al., 2016*; *Wilhelm et al., 2013*), raising the question of which features during sleep might support such high levels of memory consolidation at young age. Interestingly, in young children the development-specifically manifested SO-spindle coupling pattern was found to not yet be associated with memory consolidation while sleep spindles and SOs individually were (*Joechner et al., 2021*), implying that alternative features during sleep might compensate for less pronounced SO-spindle coupling (*Wilhelm et al., 2012*). However, these questions remain to be addressed in future longitudinal research.

## Conclusion

Overall, our findings describe a unique relation between fast sleep spindle maturity and the pattern of SO-spindle coupling across childhood, adolescence, and young adulthood. While SOs provide optimal time windows for sleep spindles to arise, it is the ability to generate adult-like, canonical fast spindles that determines coupling strength and precision across child and adolescent development. Given the evidence for its generating role, our results implicate the maturation of specific thalamocortical circuits as the cornerstone of adult-like SO-spindle coupling patterns. Hence, our findings represent a promising starting point for future research addressing the precise relation between age-related changes in brain structure and function, the emergence of adult-like SO-spindle coupling, and cognitive development across childhood and adolescence.

# Materials and methods

All analyses presented here are based on two previously published datasets: one cross-sectional cohort of 5- to 6-year-old children (*Joechner et al., 2021*) and one longitudinal sample of children, tested initially between ages 8–11 years (*Hoedlmoser et al., 2014*) and again around seven years later between ages 14–18 years (*Hahn et al., 2019*; *Hahn et al., 2020*) during adolescence. Please refer to the original studies for a detailed description of all inclusion and exclusion criteria and the experimental procedures, respectively. In addition, an unpublished cross-sectional sample of young adults aged between 20 and 26 years of age was analyzed.

## Participants

For the cross-sectional child study, originally, 36 healthy, German-speaking 5- to 6-year-old pre-school children (19 female, $M_{age}$ = 5 years, 9.53 months, $SD_{age}$ = 6.50 months) were recruited from the database of the Max Planck Institute for Human Development (MPIB), Berlin, Germany, and from daycare centers in Berlin, Germany. All 36 initially enrolled participants did not show any evidence of the use of medication, a personal or family history of mental and sleep disorders, learning disabilities, respiratory problems, and obesity. Of these, five participants did not complete the study, and another seven children were excluded due to technical issues (n = 4) during PSG or missing compliance with the study protocol (n = 3). Therefore, the same 24 5-to 6-year-old participants (13 female, $M_{age}$ = 5 years, 10.71 months, $SD_{age}$ = 7.28 months) as in *Joechner et al., 2021* are presented here.

The initial sample of the longitudinal cohort consisted of 63 healthy, Austrian children (T1, 28 female, $M_{age}$ = 10 years, 1.17 months, $SD_{age}$ = 7.97 months; *Hoedlmoser et al., 2014*), recruited from public elementary schools in Salzburg, Austria. Approximately seven years later, 36 participants

returned for a follow-up assessment (T2, 24 female, $M_{age}$ = 16 years, 4.56 months, $SD_{age}$ = 8.76 months; *Hahn et al., 2019*; *Hahn et al., 2020*) during adolescence. At T1, none of the participants showed any signs of sleep or mental disorders, medication use, respiratory problems, and obesity. Two adolescents had to be excluded due to technical problems during PSG and data of one participant was not further analyzed because of insufficient amounts of NREM sleep stage 3 (N3; out of four nights, in only two nights N3 sleep was detected; together: 3.05%). Hence, we here present repeated-measures data for a total of 33 participants (23 female, $M_{ageT1}$ = 9 years, 11.70 months, $SD_{ageT1}$ = 8.35 months; $M_{ageT2}$ = 16 years, 4.91 months, $SD_{ageT2}$ = 9.06 months).

The cross-sectional sample of adults initially included 19 healthy, Austrian participants recruited at the University of Salzburg, Austria (15 female, $M_{age}$ = 21 years, 8. 58 months, $SD_{age}$ = 19.49 months). One participant had to be excluded from analyses due to bad signal quality of the EEG data. Hence, we analyzed a final sample of 18 adult participants between the ages of 20 and 26 here (15 female, $M_{age}$ = 21 years, 8.78 months, $SD_{age}$ = 20.04 months). Minor participants and their families received a gift (5- to 6-year-old and 8- to 11-year-old participants) and/or monetary compensation (the parents of the 5- to 6-year-olds and the 14- to 18-year-olds) for their study participation. Adult participants received either monetary compensation or study credit points for their participation.

All studies were designed in accordance with the Declaration of Helsinki and approved by the local ethics committee of either the MPIB, Germany (2018_5_Sleep_Conmem), or the University of Salzburg, Austria (EK-435 GZ:16/2014). For the 5- to 6-year-olds, legal guardians gave their written and minors gave their oral informed consent prior to study participation. For the longitudinal study, participants and their legal custodian provided written informed consent before entering the study at every test time point. Similarly, adults provided written informed consent. Given that participants were assessed during four distinct age intervals, the term 'age group(s)' is used to refer to 5- to 6-, 8- to 11-, 14- to 18-, and the 20- to 26-year-olds.

## General procedure

Despite procedural differences, all three studies comprised two nights of ambulatory PSG for each test time point. While the first night served to familiarize participants with the PSG, a memory task was performed before and after the second night (an associative scene-object task for the cross-sectional child study and an associative word-pair task for both time points of the longitudinal study and the adult cross-sectional sample). Sleep was monitored in the habitual environment of each participant. For each 5- to 6-year-old participant, PSG recordings started and ended according to the individual bedtime. For the participants in the longitudinal sample, recordings were scheduled between 7:30–8:30 p.m. and 6:30 a.m. and were stopped after a maximum of 10 hr time in bed during childhood. During the follow-up assessment, time in bed was fixed to 8 hr between 11 p.m. to 7 a.m. For the cross-sectional adult sample, the procedure was exactly the same as for the longitudinal follow-up assessment. Given well-known first night effects in children (*Scholle et al., 2003*), only the second night was analyzed here.

## Sleep EEG acquisition and analyses
### Sleep recordings

For the youngest cohort, sleep was recorded using an ambulatory amplifier (SOMNOscreen plus, SOMNOmedics GmbH, Randersacker, Germany). A total of seven gold cup electrodes (Grass Technologies, Natus Europe GmbH, Planegg, Germany) were placed on the scalp for EEG recordings (F3, F4, C3, Cz, C4, Pz, Oz). EEG channels were recorded at a sampling rate of 128 Hz against the common online reference Cz. The signal of the AFz served as ground. Initial impedance values were kept below 6 kΩ. Additionally, two electrodes were placed at the left and right mastoids (A1, A2) for later re-referencing. Horizontal electrooculogram (EOG) was assessed bilaterally around the eyes and electromyogram (EMG) was recorded on the left and right musculus mentalis, referenced to one chin electrode. Furthermore, cardiac activity was monitored with two electrocardiogram (ECG) channels.

For the three older age groups, PSG signals were also collected ambulatory with a portable amplifier system (Varioport, Becker Meditec, Karlsruhe, Germany) at a sampling frequency of 512 Hz and against the common reference at Cz. For EEG recordings, 11 gold-plated electrodes (Grass Technologies, Natus Europe GmbH, Planegg, Germany; F3, Fz, F4, C3, Cz, C4, P3, Pz, P4, O1, O2) were placed

along with A1 and A2 at the bilateral mastoids for offline re-referencing. Further, two horizontal and two vertical EOG channels, as well as two submental EMG channels, were recorded.

## Pre-processing and sleep staging

Sleep was staged automatically (Somnolyzer 24x7, Koninklijke Philips N.V.; Eindhoven, The Netherlands) and visually controlled by an expert scorer according to the criteria of the American Academy of Sleep Medicine (AASM; see *Supplementary file 1w* for sleep architecture). Initial pre-processing was performed using BrainVision Analyzer 2.1 (Brain Products, Gilching, Germany). EEG channels were re-referenced offline against the average of A1 and A2 and filtered between 0.3–35 Hz. Further, all EEG data were resampled to 256 Hz. All subsequent analyses were conducted using Matlab R2016b (Mathworks Inc, Sherborn, MA) and the open-source toolbox Fieldtrip (*Oostenveld et al., 2011*). Firstly, bad EEG channels were rejected based on visual inspection. The remaining channels were then cleaned by applying an automatic artifact detection algorithm on 1 s segments (see *Joechner et al., 2021*; *Muehlroth et al., 2019* for more details).

## Sleep spindle and slow oscillation detection

Sleep spindles were detected during artifact-free NREM (N2 and N3) epochs at frontal (F3, F4) and centro-parietal channels (C3, C4, Cz, Pz) using an established algorithm (*Klinzing et al., 2016*; *Mölle et al., 2011*; *Muehlroth et al., 2019*) with individual amplitude thresholds. Frequency windows for spindle detection were defined based on two approaches: (i) An individualized approach, aiming at capturing the person- and development-specific dominant rhythm, and (ii) a fixed approach, targeting 'adult-like' fast sleep spindles in the canonical fast spindle range (12.5–16 Hz) that were shown to be coupled to SOs before, despite missing evidence for their strong presence in pre-school children (*Joechner et al., 2021*; see also *Figure 1A* and *Figure 3*). For the individualized approach, we defined the frequency bands of interest as the participant-specific peak frequency in the averaged frontal (F3, F4) or centro-parietal (C3, Cz, C4, Pz) background-corrected NREM power spectra (9–16 Hz) ± 1.5 Hz (*Mölle et al., 2011*; *Ujma et al., 2015*; see *Figure 1A* for peak distributions). Therefore, NREM power spectra were calculated within participants for averaged frontal and centro-parietal electrodes between 9–16 Hz by applying a Fast Fourier Transform (FFT) and using a Hanning taper on 5 s epochs. To restrict the search space for peak detection to dominant rhythmic activity (*Aru et al., 2015*; *Kosciessa et al., 2020*), we then modeled the background spectrum and subtracted it from the original power spectrum. Assuming that the EEG background spectrum follows an $A*f^{-a}$ distribution (*Buzsáki and Mizuseki, 2014*; *He et al., 2010*), background spectra were estimated by fitting the original power spectrum linearly in the log(power)-log(frequency) space employing robust regression (*Kosciessa et al., 2020*). Frontal and centro-parietal peaks were finally detected in the background-corrected power spectra by a classical search for maxima combined with a first derivative approach (*Grandy et al., 2013*). For the fixed approach, frequency bands were restricted to 12.5–16 Hz in every individual—representing the canonical range for fast sleep spindle extraction in adults—and detection was focused on averaged centro-parietal electrodes only to capture adult-like fast sleep spindles (C3, Cz, C4, Pz; *Cox et al., 2017*). For detection of spindle events, EEG data were first filtered in the respective frequency bands using a 6[th] order two-pass Butterworth filter (separately for individually identified frontal and centro-parietal, as well as adult-like fast centro-parietal spindles). Then, the root mean square (RMS) was calculated using a moving window of 0.2 s and smoothed with a moving average of 0.2 s. Finally, a sleep spindle was detected whenever the envelope of the smoothed RMS signal exceeded its mean by 1.5 SD of the filtered signal for 0.5–3 s. Sleep spindles with boundaries within a distance between 0–0.25 s were merged as long as the resulting event was shorter than 3 s (see, e.g., *Mölle et al., 2011*; *Muehlroth et al., 2019*, for similar methods; see *Figure 1—figure supplements 1 and 2* for the average EEG signal time-locked to all different spindle types and for examples of how individually identified (i.e., development-specific) and adult-like fast centro-parietal spindles manifested in the raw EEG).

SO detection was based on an algorithm with individually adapted amplitude thresholds (*Mölle et al., 2002*; *Muehlroth et al., 2019*) and performed for all NREM epochs. First, the EEG signal was filtered between 0.2 and 4 Hz using a two-pass Butterworth filter of 6[th] order. Subsequently, zero-crossings were marked to identify positive and negative half-waves. Pairs of negative and succeeding positive half-waves were considered a potential SO if their frequency was 0.5–1 Hz. Only putative SOs

with a peak-to-peak amplitude of 1.25 times the mean peak-to-peak amplitude of all tagged SOs and a negative amplitude of 1.25 times the average negative amplitude of all SOs that did not include artifact segments were kept for the following analyses.

## Temporal association between sleep spindles and slow oscillations

To assess the temporal association between sleep spindles and SOs, we employed two different approaches following previous reports in developmental samples (*Joechner et al., 2021*; *Muehlroth et al., 2019*).

### Time-frequency analyses

On the one hand, we examined power modulations during SOs between 5–20 Hz within intervals of ± 1.2 s around the down peak of SOs. For this, we detected artifact-free NREM epochs containing SOs (down peak ± 3 s) and equally long, randomly chosen segments without SOs. Time-frequency representations (*Figure 3*) were then calculated for trials with and without SOs using a Morlet wavelet transformation (12 cycles) in steps of 1 Hz and 0.002 s. The resulting time-frequency pattern of trials with and without SOs was contrasted for every participant using independent sample t-tests. Subsequent group-level analyses were conducted separately for the cross-sectional cohorts and the two cohorts from the longitudinal sample. Within these four age groups, the t-maps were then contrasted against zero using a cluster-based permutation test (*Maris and Oostenveld, 2007*; two-sided, critical alpha-level α=0.05; note for the ease of comparability, p-values for all cluster-based permutation tests were multiplied by 2 and thus values below 0.05 were considered significant) with 5000 permutations in a time window between –1.2 to +1.2 s around the SO down peak.

### Temporal co-occurrence of sleep spindle events with slow oscillations

On the other hand, we investigated the temporal co-occurrence of individually identified frontal and centro-parietal and adult-like fast sleep spindle events with SOs. In a first step, we calculated the general co-occurrence rate between sleep spindles and SOs on a broad time scale by determining the percentage of spindle event centers that occurred within an interval of ± 1.2 s around SO down peaks during NREM sleep, relative to all spindles during NREM sleep. We also repeated this analysis vice versa for SO down peaks occurring within an interval of ± 1.2 s around spindle centers (see *Figure 4— figure supplements 3 and 4* and *Supplementary file 1x–ac*).

To specify the temporal relation between sleep spindle and SO co-occurrence on a finer temporal scale, in a second step, we calculated PETHs (*Figure 4*). Therefore, the interval of ± 1.2 s around a SO down peak was partitioned into 100 ms bins and the proportion of sleep spindle centers occurring within each time bin was assessed. The occurrence rates within a bin were subsequently normalized by the total counts of sleep spindles occurring during the complete respective SO down peak ± 1.2 s interval and multiplied by 100. To determine whether spindle activity was modulated during SOs differently from what would be expected by chance, we created a surrogate comparison distribution for every participant during every test time point. The original percentage frequency distribution was randomly shuffled 1000 times and averaged across all permutations. The resulting surrogate distributions were then compared against the original distributions using dependent sample t-tests. To account for the multiple comparisons, a cluster-based permutation test with 5000 permutations was implemented (*Maris and Oostenveld, 2007*; two-sided, critical alpha-level α=0.05, p-values for all cluster-based permutation tests were multiplied by 2 and thus values below 0.05 were considered significant). All analyses were conducted separately for individually identified sleep spindles at frontal (averaged over F3, F4) and centro-parietal (averaged over C3, Cz, C4, Pz) electrodes and for adult-like fast spindles detected at centro-parietal (averaged over C3, Cz, C4, Pz) derivations. For a control analysis on the influence of the number of events on PETH patterns, we 100 times randomly drew half the number of adult-like fast sleep spindles within a participant and calculated co-occurrence rates. We than averaged across these 100 surrogates and recalculated the PETHs.

To quantify the modulation strength of sleep spindle occurrence during SOs, for every participant, we calculated the KL divergence between the percentage frequency distribution (p) and its surrogate (q) used for the PETHs. The KL divergence is a measure rooted in information theory that describes the amount of information loss if one distribution was approximated by the other (*Joyce, 2011*). This

measure is the basis for several commonly used methods to determine phase-amplitude coupling and was calculated in the following way:

$$\mathrm{D_{KL}(p \,||\, q)} = \sum_{i=1}^{N}(p(x_i) * \ln\tfrac{p(x_i)}{q(x_i)})$$

The higher the value, the more two distributions deviate from each other. Hence, higher values indicate that the actual percentage frequency distribution of the PETHs deviated from the surrogate distribution, that is, the more likely sleep spindles were concentrated around specific phases of the underlying SO.

## Statistical analyses

### Age differences in sleep spindles and slow oscillations

Age-related comparisons between sleep spindle and SO parameters were conducted using LMMs with restricted maximum likelihood variance estimation (REML) and the bobyqa optimizer from the NLopt library (**Powell, 2009**). LMMs were implemented in R 4.0.3 (**R Development Core Team, 2020**) using Rstudio Version 1.1.383 and the lme4 package (**Bates et al., 2015**). Given that our data consist of both, purely cross-sectional as well as repeated measures, participants were included as random effects to account for the nonindependence when dealing with repeated measures. Fixed effects were most of the time 'age group' (i.e., 5- to 6-, 8- to 11-, 14- to 18-, 20- to 26-year-olds) and 'topography' (e.g., frontal, central, occipital) and specified based on our research questions. The effects of categorical predictors were set up as sum-coded factors. Overall, only models with a random intercept for participants converged (but not with a random slope). Results were summarized by F-statistics and p-values based on Type III sums of squares estimation and the Satterthwaite's method provided by lmerTest (**Bates et al., 2015**; **Kuznetsova et al., 2017**). Post-hoc pairwise comparisons were conducted using the emmeans package (**Lenth, 2021**). Degrees of freedom (df) were calculated applying the Satterthwaite's method (**Giesbrecht and Burns, 1985**; **Kuznetsova et al., 2017**) and p-values were Bonferroni corrected ($p_{adj}$; **Bland and Altman, 1995**).

### Association between sleep spindle and slow oscillation maturity with slow oscillation-spindle coupling

To assess the association between sleep spindle and SO maturity and SO-spindle coupling (for an exact definition of the variables, see the results section), we conducted two analyses:

First, a PLSC (**Krishnan et al., 2011**) was calculated between indicators of sleep spindle amplitude, frequency, and density maturity and the t-maps of power modulations during SOs (compared to non-SO trials, time-frequency maps) across all age groups. PLS methods are multivariate tools to extract commonalities between two data sets using singular value decomposition. PLSC specifically analyzes the association between two data sets by deriving pairs of singular vectors that cover the maximal covariance between two matrices (**Krishnan et al., 2011**). Singular vector pairs are ordered by the amount of covariance they contribute to the association as reflected by their corresponding singular values. The components of the singular vectors represent weights (also called saliences) that define how each element of the data sets contributes to a given association. Ultimately, the projection of pairs of saliences onto their original data matrices results in pairs of latent variables that capture the maximal amount of common information between the two data sets (e.g., in our case a latent time-frequency and latent spindle maturity variable). Therefore, PLSC is ideally suited to identify time-frequency patterns associated with a specific spindle maturity profile. The number of impactful singular vector pairs (i.e., those that account for a significant amount of covariance) was identified by a permutation test with 5000 permutations based on the singular values. For each significant vector pair, reliability of the weights of each time-frequency value, defining the SO-spindle coupling pattern associated with a spindle maturity pattern, was determined using bootstrap ratios (BSR). BSRs represent the ratio of the weights of each time-frequency value and their bootstrap standard errors based on 5000 samples. BSRs are akin to Z-scores, hence BSRs higher than 1.96 and lower than –1.96 were considered stable (**Krishnan et al., 2011**). The spindle maturity profile was represented as the correlation between the latent time-frequency variable and the three raw spindle maturity variables. These values are comparable to the spindle maturity weights and indicate whether the association between the time-frequency pattern and individual spindle maturity variables is in the

same or different direction (*McIntosh and Lobaugh, 2004*). Stability of these correlation patterns was determined by their bootstrap estimated 95% confidence intervals (*McIntosh and Lobaugh, 2004*). Note, despite our awareness of different dependencies between the age groups, 5- to 6-, 8- to 11-, 14- to 18-, and 20- to 26-year-olds were considered for this analysis.

Second, GLMMs were used to examine the relation between the modulation of sleep spindles during SOs (as captured by the KL divergence of the PETHs) with spindle and SO maturity across all age groups. We defined the first component of a PCA between indicators of sleep spindle amplitude, frequency, and density maturity across the complete data set (across all age groups) as a general spindle maturity factor. We then aimed to associate the KL divergence values, reflecting sleep spindle modulation strength during SOs, with the general spindle maturity component. For this, we set up log-linked gamma GLMMs with the KL divergence as the dependent variable and with the spindle maturity component as a fixed factor, allowing for random intercepts per participant. In addition, an indicator of SO maturity was added as a covariate. We used the bobyqa optimizer from the minqa package (*Bates et al., 2014*).

## Acknowledgements

This research was conducted within the project "Lifespan Rhythms of Memory and Cognition" (RHYME, PI: MW-B) at the Max Planck Institute for Human Development (MPIB), Berlin, Germany, and at the Department of Psychology, Laboratory for Sleep, Cognition and Consciousness Research, University of Salzburg, Salzburg, Austria (PI: KH). A-KJ is a fellow of the International Max Planck Research School on the Life Course (LIFE; http://www.imprs-life.mpg.de/en). MW-B received support from the German Research Foundation (DFG, WE 4269/5–1) and the Jacobs Foundation (Early Career Research Fellowship 2017–2019). KH was supported by Austrian Science Fund (T397-B02, P25000), the Jacobs Foundation (JS1112H), and the Centre for Cognitive Neuroscience Salzburg (CCNS). MAH was additionally supported by the Doctoral College "Imaging the Mind" (FWF, Austrian Science Fund W1233-G17) and the PRIME programme of the German Academic Exchange Service (DAAD) with funds from the German Federal Ministry of Education and Research (BMBF). We thank S Wehmeier for her invaluable help with data collection at the MPIB. We are grateful to all members of the RHYME and LIME projects for valuable feedback. We further acknowledge support by the Max Planck Dahlem Campus of Cognition (MPDCC). Finally, we thank all our participants and their families for their time as well as the principals of the schools and the local education authority (Mag. Dipl. Paed. B Heinrich, Prof. Mag. J Thurner) in Salzburg who supported this research.

## Additional information

### Funding

| Funder | Grant reference number | Author |
| --- | --- | --- |
| Deutsche Forschungsgemeinschaft | WE 4269/5-1 | Markus Werkle-Bergner |
| Jacobs Foundation | Early Career Research Fellowship 2017-2019 | Markus Werkle-Bergner |
| Austrian Science Fund | T397-B02 | Kerstin Hoedlmoser |
| Jacobs Foundation | JS1112H | Kerstin Hoedlmoser |
| Austrian Science Fund | W1233-G17 | Michael A Hahn |
| Austrian Science Fund | P25000 | Kerstin Hoedlmoser |
| German Academic Exchange Service | PRIME Fellowship | Michael A Hahn |

The funders had no role in study design, data collection and interpretation, or the decision to submit the work for publication. Open access funding provided by Max Planck Society.

### Author contributions
Ann-Kathrin Joechner, Conceptualization, Data curation, Software, Formal analysis, Methodology, Writing – original draft, Project administration, Writing – review and editing; Michael A Hahn, Kerstin Hoedlmoser, Conceptualization, Data curation, Writing – review and editing; Georg Gruber, Software, Writing – review and editing; Markus Werkle-Bergner, Conceptualization, Data curation, Funding acquisition, Methodology, Project administration, Writing – review and editing

### Author ORCIDs
Ann-Kathrin Joechner ⓘ https://orcid.org/0000-0003-4962-1089
Michael A Hahn ⓘ http://orcid.org/0000-0002-3022-0552
Kerstin Hoedlmoser ⓘ http://orcid.org/0000-0001-5177-4389
Markus Werkle-Bergner ⓘ http://orcid.org/0000-0002-6399-9996

### Ethics
All studies were designed in accordance with the Declaration of Helsinki and approved by the local ethics committee of either the MPIB, Germany (2018_5_Sleep_Conmem), or the University of Salzburg, Austria (EK-435 GZ:16/2014). For the 5- to 6-year-olds, legal guardians gave their written and minors gave their oral informed consent prior to study participation. For the longitudinal study, participants and their legal custodian provided written informed consent before entering the study at every test time point. Similarly, adults provided written informed consent.

### Decision letter and Author response
Decision letter https://doi.org/10.7554/eLife.83565.sa1
Author response https://doi.org/10.7554/eLife.83565.sa2

---

## Additional files

### Supplementary files
• Supplementary file 1. Complete record of all statistical results and analyses. (a) Type III analysis of variance table from a linear mixed-effects model on the effects of age group and sleep spindle topography on sleep spindle spectral peak frequency. (b) (A) Post-hoc comparisons for the topography effect on sleep spindle peak frequency based on estimated marginal means (B) Post-hoc comparisons for the age group effect on sleep spindle peak frequency based on estimated marginal means. (c) Type III analysis of variance table from a linear mixed-effects model on the effects of age group and sleep spindle topography on sleep spindle frequency. (d) Post-hoc comparisons for the sleep spindle frequency interaction effect based on estimated marginal means. (e) Type III analysis of variance table from a linear mixed-effects model on the effects of age group and sleep spindle topography on sleep spindle density. (f) Post-hoc comparisons for the sleep spindle density interaction effect based on estimated marginal means. (g) Type III analysis of variance table from a linear mixed-effects model on the effects of age group and sleep spindle topography on sleep spindle amplitude. (h) Post-hoc comparisons for the sleep spindle amplitude interaction effect based on estimated marginal means. (i) Type III analysis of variance table from a linear mixed-effects model on the effects of age group and sleep spindle type (development-specific and adult-like fast sleep spindles) on frequency. (j) Post-hoc comparisons for development-specific and adult-like fast sleep spindle frequency values within the four age groups (interaction effect) based on estimated marginal means. (k) Type III analysis of variance table from a linear mixed-effects model on the effects of age group and sleep spindle type (development-specific and adult-like fast sleep spindles) on density. (l) Post-hoc comparisons for development-specific and adult-like fast sleep spindle density within the four age groups (interaction effect) based on estimated marginal means. (m) Type III analysis of variance table from a linear mixed-effects model on the effects of age group and sleep spindle type (development-specific and adult-like fast sleep spindles) on amplitude. (n) Post-hoc comparisons for development-specific and adult-like fast sleep spindle amplitude within the four age groups (interaction effect) based on estimated marginal means. (o) Type III analysis of variance table from a linear mixed-effects model on the effects of age group and slow oscillation topography on slow oscillation frequency. (p) Post-hoc comparisons for the slow oscillation frequency interaction effect based on estimated marginal means. (q) Type III analysis of variance table from a linear mixed-effects model on the effects of age group and slow oscillation topography on slow oscillation density. (r) Post-hoc comparisons for the effect of age group on slow oscillation density values based on estimated marginal means. (s) Type III analysis of

variance table from a linear mixed-effects model on the effects of age group and slow oscillation topography on slow oscillation amplitude. (t) Post-hoc comparisons for the slow oscillation amplitude interaction effect based on estimated marginal means. (u) Results of the generalized linear mixed-effects model on the association between the fast sleep spindle maturity component and frontal slow oscillation-development-specific fast spindle modulation strength. (v) Results of the generalized linear mixed-effects model on the association between the fast sleep spindle maturity component and (A) centro-parietal and (B) frontal slow oscillation-development-specific fast spindle modulation strength while also controlling for age. (w) Sleep architecture. (x) Type III analysis of variance table from a linear mixed-effects model on the effects of age group and slow oscillation topography on the co-occurrence of slow frontal sleep spindles with slow oscillations. (y) Type III analysis of variance table from a linear mixed-effects model on the effects of age group and slow oscillation topography on the co-occurrence of development-specific fast centro-parietal sleep spindles with slow oscillations. (z) Type III analysis of variance table from a linear mixed-effects model on the effects of age group and slow oscillation topography on the co-occurrence of slow oscillations with slow frontal sleep spindles. (aa) Type III analysis of variance table from a linear mixed-effects model on the effects of age group and slow oscillation topography on the co-occurrence of slow oscillations with development-specific fast centro-parietal sleep spindles. (ab) Type III analysis of variance table from a linear mixed-effects model on the effects of age group and slow oscillation topography on the co-occurrence of adult-like fast centro-parietal sleep spindles with slow oscillations. (ac) Type III analysis of variance table from a linear mixed-effects model on the effects of age group and slow oscillation topography on the co-occurrence of slow oscillations with adult-like fast centro-parietal sleep spindles.

- MDAR checklist

## Data availability

All data and code to reproduce the present analyses and result figures are publicly available through the Open Science Framework (https://osf.io/2u6ec/).

The following dataset was generated:

| Author(s) | Year | Dataset title | Dataset URL | Database and Identifier |
|---|---|---|---|---|
| Joechner A-K, Werkle-Bergner M | 2022 | Sleep spindle maturity promotes slow oscillation-spindle coupling across child and adolescent development | https://osf.io/2u6ec/ | Open Science Framework, 2u6ec |

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
