## [Editor Report]

This is an important analysis of sleep datasets across different age groups that contributes to our understanding of sleep spindle and slow oscillation dynamics during development. The work is expected to be of interest to interdisciplinary fields including development and sleep. The analyses are solid and adequately complex to capture the changes in sleep spindle to slow oscillation coupling among the age groups.

---

## [Decision Letter]

**Decision letter after peer review:**

Thank you for submitting your article "Sleep spindle maturation enhances slow oscillation-spindle coupling" for consideration by *eLife*. Your article has been reviewed by 3 peer reviewers, and the evaluation has been overseen by Laura Colgin as the Senior Editor. The following individual involved in review of your submission has agreed to reveal their identity: Hong-Viet Ngo (Reviewer #3).

Essential revisions:

1) A major limitation of the work is that analogous analyses were not performed on sleep data recorded from adult subjects. Reviewers strongly encourage including results from adults in the resubmitted manuscript. However, if it is not feasible to complete such analyses within the time allowed for revisions, the expectation would be that supporting data from adults would be submitted to *eLife* in the future as a follow-up to the present study (e.g., submit the new results from adults to *eLife* as a Research Advance in the future see: https://reviewer.elifesciences.org/author-guide/types.

2) Additional analyses and revisions suggested to improve clarity and maximize impact of the work are provided in the individual reviews below.

*Reviewer #1 (Recommendations for the authors):*

1. An unfortunate and I think unnecessary limitation of this manuscript is that a sample of adults is missing. This would be highly desirable to show how the very complex analyses and transformations compare to previously reported analyses in young healthy adults. It would make interpreting the data much easier and there are plenty such samples available either in the author's own labs or from their collaborators throughout Germany, Austria and Switzerland. I would urge the authors to add such data to the manuscript.

2. One other concern regards the spindle maturation index. The authors report two fast spindle peaks: one age adapted and one canonical. What is the evidence that the "age-adapted" fast spindles are indeed fast spindles? How do we know that they are generated by the same thalamocortical circuits? I believe this is crucial for the argument of the authors. Since the "age-adapted" fast spindles are not modulated by the SO this may indicate that they are not the same phenomenon. Clearly, this cannot be shown conclusively in this paper, but I would urge the authors to more stringently test this, to then thoroughly discuss this and to compare the features of these oscillations more thoroughly.

3. Since the second time point in the longitudinal sample was assessed during puberty, I was wondering, if the authors took sex into account. It seems likely that sex hormones could affect their findings.

4. Since the authors find less canonical spindles in younger participants, could the effect of imprecise spindle to SO coupling shown in Figure 4 be due to noise? To check this the authors could randomly choose a subset of spindles for the adolescents and see how precision compares then.

5. Related to this. The findings that spindle maturation is related to SO-spindle coupling seems like double dipping. First, the authors identify age related differences in two measures and then they find a relationship between them. I think it would be a stronger finding, if this relationship was shown within each age group individually. However, I may have misunderstood this approach.

6. Generally, the analyses are a bit hard to understand, since for example a lot of transformations are being performed before analysis. I would suggest adding some more descriptive figures to the manuscript that allow judging the pipeline from raw data to inference. For example, I would like to see averages (for all subjects) and grand averages (per age group) of the different types of detected spindles. I would also like to see the spindle band-filtered EEG signal (for each type of spindle) time-locked to the slow-oscillation down-state. The authors should also add figures showing the averaged (and SD) spectral analyses that were used to identify the different spindle peaks. Also the individual spectra could be shown larger in the supplement.

7. Can the authors analyse the phase-amplitude-coupling of the SOs and canonical spindles per age group to extract the phase angle? There seems to be a phase progression from younger to older, which I find quite interesting.

8. The findings are of course the result of data exploration. This should be clearly acknowledged. It might be helpful to add some robustness analyses to show that the findings are not due to specific choices made during the exploration process.

9. The time-frequency plots time-locked to SOs do not seem to show a meaningful comparison. This is usually done by comparing two conditions rather than to a baseline. I am not sure we can learn much from them, if there is just a huge cluster across the whole spectrum and time.

10. For better readability: please do not abbreviate spindles.

Typos: "However, in neither age group, we did" (also: AFAIK "neither" is only used for two alternatives).

*Reviewer #2 (Recommendations for the authors):*

P3. Line 5: memories are not "transferred" to the cortex but strengthened in the cortex (since already encoded in the cortex).

After reading the whole article it is clear why the authors use "canonical" for the "normal" spindles, but it would be easier for readers if the authors the first time when they use the word, explain why and define it.

*Reviewer #3 (Recommendations for the authors):*

Title: The title is rather unspecific with respect the age range and could be specified further.

Figures: Almost all plots showing individual data points, means and raincloud plots are very hard to interpret and could need a small overhaul. In particular Figure 2, does not allow a visual assessment of the differences between cortical areas.

Page 1, line 4: For consistency, the authors could also add the frequency range of SOs here.

Page 2, line 11: Here it is stated that how the coupling develops is an open question. However, at the end of the next paragraph, a previous publication is reported that shows how the coupling improves across childhood. Is this open question still valid then?

Moreover, throughout the introduction, the authors repeat this open question several times with slights variations in the phrasing. I wonder if this is necessary or if the open question can be asked centrally before the final paragraph of the introduction.

Page 6, line 4: Please see my comment in the public review. I think the assessment of average frequency of spindle events is redundant or even circular as they are based on the power spectra.

Page 6, line 18: The introduction of development-specific feels unnecessary and to some degree arbitrary. At least for me, it only led to overly complicated/cluttered sentences. Is this term really required or can it be omitted while the main message is conveyed properly.

Page 10, Figure 1: It would help the reader if the age ranges are illustrated in subpanel A. Moreover, as stated above, symbols for the different spindle types and the marginal means are very hard to distinguish. Also, I was at first not aware that the adult-like spindles even had a rain cloud. Perhaps the authors can rearrange/resize the figure.

Related to subpanel A, its known that adults sometimes do not show slow spindle peaks or shows both a slow and fast spindle peak at one electrode (e.g., Mölle et al., 2011). At least the former seems to be the same here. How did the authors proceed if that was the case? I was wondering if it would be more informative to sort the individual subjects peak frequency per age group.

Page 14, line 4: What do the authors mean by "more precise"? The size of the cluster or its location with respect to the SO up-state? Is it really possible to make conclusion about the precision using a TFR-based analysis affected by temporal and frequency smearing?

Page 14, Figure 2. A comparison between SO vs. SO-free intervals includes a strong difference in overall spectral power, hence the wide significant cluster. The authors could consider decreasing their α-value further to tease out clusters more clearly. Furthermore, the dashed window highlighting the spindle range is not really visible.

Page 16, Figure 4: Again, this is reflecting my personal opinion, but this the central result and it somehow feels like it does not get enough attention. Anyhow, what does the horizontal jiggly line represent? It is also stated that the empirical PETHs are compared to a surrogate distribution, however, is that really necessary? The authors have SO-free intervals that could be used for a cluster-based permutation test as well.

Page 19, Figure 6: Can the author please explain what the main message of subpanel A is? This somehow eludes me. Also, how come there is not separation into the different age groups?

---

## [Author Response]

Essential revisions:Reviewer #1 (Recommendations for the authors):1. An unfortunate and I think unnecessary limitation of this manuscript is that a sample of adults is missing. This would be highly desirable to show how the very complex analyses and transformations compare to previously reported analyses in young healthy adults. It would make interpreting the data much easier and there are plenty such samples available either in the author's own labs or from their collaborators throughout Germany, Austria and Switzerland. I would urge the authors to add such data to the manuscript.

We now included the analysis of a sample of adults that were recorded with the same ambulatory amplifier and setup as the longitudinal sample.

In summary, the analyses on the adult age group showed similar results as in the age group of 14- to 18-year-olds. For sleep spindle analyses, in line with the literature (Cox et al., 2017; Ujma et al., 2015), the comparison between individually identified centro-parietal spindles (in the child age groups, development-specifically manifested) and those detected based on fixed canonical criteria (12.5–16 Hz) nicely demonstrates that there were no differences in any of the assessed features, making us even more confident to call these sleep spindles adult-like fast sleep spindles. Slow oscillation-spindle co-occurrence analyses revealed the expected pattern of increased occurrence of fast sleep spindle activity in the canonical frequency range (≈ 12.5–16 Hz) during the slow oscillation up peak and decreased activity during the down peak, highly similar to the results from the 14- to 18-year-olds. Lastly, taking the adult sample into account for the analyses on how sleep spindle and slow oscillation maturity relate to slow oscillation-spindle coupling patterns across all age groups revealed comparable results as before. Namely that a higher similarity between dominant, development-specific fast sleep spindles and adult-like canonical fast spindle features was associated with a more pronounced slow oscillation-spindle coupling pattern.

2. One other concern regards the spindle maturation index. The authors report two fast spindle peaks: one age adapted and one canonical. What is the evidence that the "age-adapted" fast spindles are indeed fast spindles? How do we know that they are generated by the same thalamocortical circuits? I believe this is crucial for the argument of the authors. Since the "age-adapted" fast spindles are not modulated by the SO this may indicate that they are not the same phenomenon. Clearly, this cannot be shown conclusively in this paper, but I would urge the authors to more stringently test this, to then thoroughly discuss this and to compare the features of these oscillations more thoroughly.

The question on the nature of distinct surface representations of sleep spindles is in our view one of the crucial questions in the field and also the central question arising from our results. Therefore, we already discuss this on page 28, lines 2–5.

To clarify, within a given participant we did not find separable “fast sleep spindle peaks” for development-specific and canonical fast spindle ranges, respectively. On the contrary, the (only) spectral peak for sleep spindles in centro-parietal areas (where fast sleep spindles usually dominate) was markedly below the frequency of canonical fast sleep spindles for the majority of younger children (see Figure 1). As described under 2.1 “Dominant fast cento-parietal sleep spindles become more prevalent and increasingly resemble canonical fast sleep spindles with older age”, we considered sleep spindles in centro-parietal areas, detected based on the individual peak frequency, as fast based on a combination of frequency (faster than dominant frontal sleep spindles) and topography (centro-parietal derivations). Traditionally, Gibbs and Gibbs (1950) have defined slow and fast spindles based on these criteria and such a definition is also commonly employed nowadays (Anderer et al., 2001; Cox et al., 2017; Mölle et al., 2011; Ujma et al., 2015; for review see: De Gennaro and Ferrara, 2003; Fernandez and Lüthi, 2019). Since we define slow frontal and (development-specific) fast sleep spindles based on the respective peak frequency within a participant, we are confident to capture comparable phenomena with this approach, namely the dominant slow and fast spindle types within a participant at a given age. This is in our view a crucial feature for age-fair comparisons. However, even in the absence of a peak in the canonical fast spindle frequency range (as for the younger children, while for older children, adolescents, and young adults, the individual centroparietal peak frequency was already within or very close to the adult-like, canonical fast spindle frequency range), we could additionally find a low number of adult-like, canonical fast sleep spindles—based on detection of individual oscillatory events in the canonical fast spindle range. While the sleep spindles detected based on the peak frequency and those based on the fixed, adult-like, canonical fast frequency range likely capture similar events for older children, adolescents, and young adults, it remains to be elucidated whether the dominant, development-specific fast spindles in young children are a slower version of canonical fast spindles, or whether they represent distinct phenomenon. As discussed on page 28, specifically the question that needs to be answered is:

“…whether development-specific and canonical fast sleep spindles represent distinct representations of the same though developing generating network or whether they originate through different structures and connections.”.

While we agree on the importance of this question, we do not see a possibility how to test this further than already done with the present data. In our view, testing the question of whether development-specific and adult-like fast sleep spindles arise from similar or distinct structures and networks would at least require high density polysomnography for source reconstructions or combined sleep polysomnography and structural magnetic resonance data in human participants—ideally in longitudinal studies.

3. Since the second time point in the longitudinal sample was assessed during puberty, I was wondering, if the authors took sex into account. It seems likely that sex hormones could affect their findings.

We agree with the reviewer that this is an important question. Indeed, sex hormones could potentially affect sleep-specific rhythmic neural activity during maturation. However, this was not a question in the context of the present study. Hence, we did not explicitly collect any information on sex steroids like e.g., estrogens and androgens. Sex differences in sleep have often been reported and associated with a variety of potential hormones like estradiol, melatonin, growth hormone and prolactin release (Franco et al., 2020; Mong and Cusmano, 2016; for review see, e.g., Attarian and Viola-Saltzman, 2020; Fernandez and Lüthi, 2020). However, the research on sexual dimorphism in sleep is still in its infancy (Mong and Cusmano, 2016) and sex differences do not (yet) allow inferences on which hormones could account for given differences. This applies even more for sexual dimorphism across maturation, where the current literature is very inconsistent (Franco et al., 2020). Since we did not have a specific hypothesis about sex differences in the context of the present study, we did not take sex into account in our analyses. Based on the previous points and given an unbalanced number of males and females in the longitudinal sample (f = 23, m = 10) and the newly added adult sample (f = 15, m=3), that we believe would bias statistical results, we would like to remain with the decision to not analyze sex differences in the context of the present paper.

However, the reviewer raises an important point. Therefore, we screened for sex differences in the course of the revision. We did not observe marked differences between females and males for our reported results (see Author response image 1 and Author response image 2 for sleep spindle and slow oscillation features grouped by sex as examples).

**Author response image 1. sa2fig1:** Sleep spindle (**A**) frequency (**B**) density, and (**C**) amplitude for slow frontal, development specific fast centro-parietal, and adult-like fast sleep spindles grouped by age group and sex. Asterisks represent the mean. Individual data points represent individual participants. Note, specifically for the adolescent and the adult sample, the number of females and males was highly unbalanced.

**Author response image 2. sa2fig2:** Slow oscillation (**A**) frequency (**B**) density, and (**C**) amplitude grouped by age group and sex. Asterisks represent the mean. Individual data points represent individual participants. Note, specifically for the adolescent and the adult sample, the number of females and males was highly unbalanced.

4. Since the authors find less canonical spindles in younger participants, could the effect of imprecise spindle to SO coupling shown in Figure 4 be due to noise? To check this the authors could randomly choose a subset of spindles for the adolescents and see how precision compares then.

Thank you for this question. Indeed, a differing total number of detected events (sleep spindle and slow oscillation events) could bias certain coupling and co-occurrence analyses. Being aware of this issue in between-person and age-comparative analyses, we already normalized occurrence rates of spindles within 100 ms time bins during a slow oscillation by the overall number of sleep spindles co-occurring with that slow oscillation per participant when creating the peri-event time histograms (PETH) instead of using absolute values. Further, we also discuss whether a lower number of sleep spindles might drive age differences in coupling patterns on page 29, lines 15–22:

“However, development-specific fast sleep spindles were not only lower in frequency but also occurred less often in children. Surprisingly though, the modulation of adult-like fast sleep spindles during SOs was well detectable across all age groups with this co-occurrence pattern becoming more precisely timed with SO up and down peaks with older age—despite huge age-related differences in their occurrence rate and the fact that adult-like fast sleep spindles showed very low density in both child cohorts (see Figure 1). Therefore, our data suggest that neither frequency nor density alone explains SO-spindle coupling patterns.”

A lower base number of spindles should only affect our PETH analyses if the selected subset of spindles would be biased on a temporal scale, but not so much if missing at random. In order to back this up, we actually ran the PETH analyses for adolescents also based on the suggested procedure: for every participant, we randomly selected half the number of detected adult-like fast sleep spindles 100 times and calculated the occurrence rates per time bin. We than averaged across these 100 surrogates and recalculated the PETHs. We observed a very similar pattern for the analyses with half the number of adult-like fast spindles as compared to the original analyses. Hence, we are quite convinced that the age-related differences in our PETH analyses are not based on noise caused by differing numbers of events. We have added this control analysis as Figure 4—figure supplement 2.

Further, it should be noted that while there was a significantly lower number of both dominant, development-specific and adult-like fast sleep spindles in younger compared to older participants, interestingly, when looking at how many percent of those co-occurred with slow oscillations, there were only slight differences in the percentage of fast sleep spindles that did occur within the time-window of a slow oscillation between the age groups. These differences were not consistent in terms of higher co-occurrence rates for older than younger participants (or vice versa), but differed depending on the slow oscillation topography and sleep spindle type, e.g., for adult-like fast sleep spindles, the 5- to 6-yearolds showed the highest slow oscillation-spindle co-occurrence with centro-parietal slow oscillations, while there were no differences between age groups in co-occurrence rates for development-specific fast spindles with frontal and centro-parietal slow oscillations (see Supplementary file 1 and Figure 4—figure supplement 3 and Figure 4—figure supplement 4). Hence, the number of events does not easily map on their temporal occurrence.

5. Related to this. The findings that spindle maturation is related to SO-spindle coupling seems like double dipping. First, the authors identify age related differences in two measures and then they find a relationship between them. I think it would be a stronger finding, if this relationship was shown within each age group individually. However, I may have misunderstood this approach.

We thank Reviewer 1 for sharing her/his concerns about associating age-related features in sleep spindles and slow oscillations with our two measures of slow oscillation-spindle coupling. Since we are not sure, whether we understood these concerns correctly, we firstly want to clarify our approach:

Within every participant, we detected fast sleep spindles based on the individual peak frequency (development-specific) and additionally using a fixed frequency band in the canonical, i.e., adult-like, fast sleep spindle frequency range (12.5–16 Hz). Since the differences between sleep spindles detected with these two methods were bigger in younger and smaller in older participants, we quantified this observation by computing the distance between these sleep spindles (e.g., average frequency of adult-like fast sleep spindles minus average frequency of development-specific fast centro-parietal sleep spindles, average amplitude of adult-like fast sleep spindles minus average amplitude of development-specific fast centro-parietal sleep spindles, average density of adult-like fast sleep spindles minus average density of development-specific fast centro-parietal sleep spindles ). As a result, for every participant within each age group, we received a numerical value that describes how close adult-like fast and dominant development-specific fast sleep spindles are within a participant at a given age. This within-participant difference measure varied with age in a way that the differences were smaller at older ages—therefore we called these difference scores fast sleep spindle maturity scores (see also Figure 5). We then aggregated these values for frequency, amplitude, and density using Principal Component Analysis which gave us what we call the fast sleep spindle maturity component (see Figure 5). In a similar way, we quantified the maturity of slow oscillations within every participant within each age group by the (dis-)similarity between frontal and centro-parietal amplitude. This within-participant difference score also varied with age in way that older participants showed a higher difference score (see Figure 5). Therefore, we termed it slow oscillation maturity score. To sum up, the maturity scores reflect features within a participant at a given age that each varied with age.

We then related these maturity scores with a quantification of the slow oscillation-spindle coupling pattern per participant at a given age (see Figure 5). Hence, we related within participant measures that presumably reflect the maturity of sleep spindles, slow oscillations, and their coupling. Importantly, while it might be possible that measures that individually are related to age in a comparable fashion could also be associated, this does not automatically imply that these measures also covary.

Our finding of a significant association between the sleep spindle maturity score(s) and the coupling patterns across our whole sample provides evidence that these features indeed covaried in our sample. This implies that there might be a relation between the manifestation of sleep spindles and the pattern of slow oscillation-spindle coupling across our studied age range.

Further, the covariation suggests that there could also be a common source of this variance related to the age of participants. Importantly though, including both sleep spindle and slow oscillation maturity measures in the generalized linear mixed-effects models did only reveal a significant relation between our measures of sleep spindle maturity and modulation strength but not with slow oscillation maturity, already indicating that age is not the only underlying source driving this relation.

As already discussed in Lindenberger and Pötter (1998), a relation between two age-related variables would be spurious if this relation is exclusively driven by age, meaning that the related variables do not share any other variance over and above that shared with age—they are completely uncorrelated when controlling for age (see Lindenbeger and Pötter, 1998, Figure 1). However, if a relation between these variables remains after partializing out age, one can be confident that this relation is influenced by common variance that is orthogonal to that of age (Lindenberger and Pötter, 1998, Figure 2). Indeed, taking age into account in our generalized linear mixed-effects models, the results reported in the manuscript and Supplementary file 1 did hold. Sleep spindle maturity was still significantly related to centroparietal (β = 0.25, *t* = 3.00, *p* = .003) and frontal slow oscillation-spindle modulation strength (β = 0.25, *t* = 3.11, *p* = .002), while the slow oscillation maturity score was not (centro-parietal slow oscillation-spindle modulation strength: β = 0.002, *t* = 0.03, *p* = .977; frontal slow oscillation-spindle modulation strength: β = 0.04, *t* = 0.64, *p* = .526). In addition, age was related to centro-parietal (β = .32, *t* = 3.76, *p* < .001) and frontal slow oscillation-spindle modulation strength (β = .18, *t* = 2.23, *p* = .026). Overall, this implies that the relation between sleep spindle maturity and slow oscillation-spindle coupling strength is not merely driven by age.

However, given its cross-sectional nature our results do neither mean that the presented pattern would hold for within-person changes across the studied age range (Lindenberger et al., 2011), nor do they imply any causality, e.g., of sleep spindle changes causing changes in coupling patterns.

Further, analyzing the relationship between spindle and slow oscillation maturity and the coupling patterns within each age group could theoretically be done. However, the following points have to be considered:

1) given the small sample sizes within each age group statistical power would be decreased

2) given the different age ranges variances likely differs between age groups

and connected to the previous points, one would assume that there is measurement invariance between the different age groups, i.e., the association is supposed to be exactly the same in 5- to 6-year-olds as in 14- to 18-year-olds which is likely not the case.

3) Therefore, analyses within age groups are subject to several influences that also bias results.

Hence, overall, we are confident that our approach is a valid and justified way to examine whether sleep spindle and slow oscillation features across different ages are related to their coupling patterns. We have added the analyses controlling for age to our Supplementary file 1 as Supplementary file 1.

6. Generally, the analyses are a bit hard to understand, since for example a lot of transformations are being performed before analysis. I would suggest adding some more descriptive figures to the manuscript that allow judging the pipeline from raw data to inference. For example, I would like to see averages (for all subjects) and grand averages (per age group) of the different types of detected spindles. I would also like to see the spindle band-filtered EEG signal (for each type of spindle) time-locked to the slow-oscillation down-state. The authors should also add figures showing the averaged (and SD) spectral analyses that were used to identify the different spindle peaks. Also the individual spectra could be shown larger in the supplement.

We thank Reviewer 1 for her/his suggestions and agree, that visualizing the different sleep spindle types is important. We might not have stated this clearly enough in the original manuscript:

“[…] see Supplementary Figure 1 for an example raw EEG trace with development specific and adult-like fast spindles and Supplementary Figure 2 for examples of average detected events”, but the original Supplementary file already included the averaged electroencephalographic (EEG) signal time-locked to the center point of (1) individually identified development-specific fast centro-parietal and (2) adult-like fast sleep spindles for exemplary participants of all age groups (now Figure 1—figure supplement 1).

This nicely illustrates that indeed the averaged EEG signal does look like a sleep spindle for both fast sleep spindle types. Further, it shows that for 5- to 6 and 8- to 11-year-olds development-specific and adult-like fast sleep spindles differ in their appearance while they appear very similar in the older age groups. Lastly, by eyeballing the signal during the different sleep spindle types, it also alludes to a link between the adult-like fast sleep spindles and slow oscillations in the 5- to 6- and the 8- to 11-year-olds.

We have now added examples for slow frontal sleep spindles (see Figure 1—figure supplement 1) and revised our figure description and how we refer to this in the main text, e.g.:

Page 5, lines 13–17: “Based on the individual frontal and centro-parietal spindle peak frequencies, we then detected sleep spindle events in frontal and centro-parietal sites which represent the person- and age-specific dominant spindle rhythms per individual (see Figure 1—figure supplement 1 for examples of averaged EEG signals time-locked to the occurrence of these dominant sleep spindles).”

Page 8, lines 8–13: “Therefore, we additionally extracted fast spindles in centro-parietal electrodes by applying fixed frequency criteria between 12.5–16 Hz (henceforth, adult-like fast sleep spindles; see Figure 1—figure supplement 2 for an example raw EEG trace with development-specific and adult-like fast spindles and Figure 1—figure supplement 1 for examples of averaged EEG signals time-locked to development-specific and adult-like fast sleep spindles).”

Similarly, a visualization of the averaged background-corrected power spectra that were used to identify the peak frequencies in frontal and centro-parietal derivations was already depicted in Figure 1. We have changed the potentially misleading titles (see Figure 1) and tried to increase the figure size and hope this increases the figure’s informative value.

Lastly, information on sleep spindle activity, time-locked to the slow oscillation down peak is reflected in Figure 3.

We hope we could highlight and clarify where to find the requested descriptive information within the presented data and hope this helps judging the data.

7. Can the authors analyse the phase-amplitude-coupling of the SOs and canonical spindles per age group to extract the phase angle? There seems to be a phase progression from younger to older, which I find quite interesting.

We thank Reviewer 1 for this comment and agree that further specification of the maturational changes in slow oscillation-spindle coupling are an exciting avenue. Theoretically, we could in addition to the time-frequency analyses (TFA) and the peri-event time histograms (PETH) also calculate phase-amplitude coupling measures on our data. However, phase-amplitude coupling measures come with a lot of pitfalls (Aru et al., 2015). In order to allow meaningful physiological interpretation, specifically in the context of age- comparative research, the explanatory power of each component has to be thoroughly established prior to phase-amplitude coupling analyses, e.g., by providing evidence for the existence of rhythmic neural activity at the time point of phase/amplitude extraction, by controlling for the confounding influence of differences in strength and frequency of rhythmic neural events, etc. (see e.g., Hahn et al., 2020; Helfrich et al., 2018). Both our employed measures for slow oscillation-spindle coupling (TFA, PETH) are tailored towards overcoming these drawbacks. The contrast of comparing power during trials with and without slow oscillations controls for power differences between the age groups. The PETHs are constructed after detection of oscillatory events, like sleep spindles and slow oscillation, and thus the caveats that apply to phase and amplitude measures do not affect this method. In addition, with the implemented randomization-test procedure (deviation from randomly shuffled distribution), we are able to assess the temporal reliability and specificity of the results contained in the PETHs. Hence, in our view the employed approaches already provide well-controlled and intuitive results on the temporal interplay between slow oscillations and sleep spindles. Admittedly, while they provide a visualization of the coupling, they do not provide a quantification. However, additional analyses in the form of phase-amplitude coupling would in our view increase the analytic complexity, but not add further information beyond what can be seen with more intuitive and simple approaches like TFAs and PETHs. Since Reviewer 3 also noted that we already report too much data, we would like to stay only with the reported analyses on SO-spindle coupling for this paper.

8. The findings are of course the result of data exploration. This should be clearly acknowledged. It might be helpful to add some robustness analyses to show that the findings are not due to specific choices made during the exploration process.

Thank you for this suggestion. Actually, as stated in the Results section (page 8, lines 13– 16) and the method section (page 50, lines 5–10) not all our analyses were purely based on a data-driven approach, but the main idea that the increased presence of adult-like fast spindles could be associated with the development of slow oscillation-spindle coupling patterns was developed based on the publication including the 5- to 6-year-olds (Joechner et al., 2021). Based on these findings, we decided to extract sleep spindles based on individual peak frequencies and also based on fixed adult-like criteria in the first place. However, we agree that we have not made clear enough in the manuscript which parts were based on ad-hoc data exploration and which were predetermined. We adjusted several lines in the manuscript to make it more transparent when we used a purely data driven approach e.g.,

Page 4, lines 14–16: “Specifically, based on previous analyses (Joechner et al., 2021), we reasoned that the development of fast sleep spindles might be associated with the maturation of SO-spindle coupling.”

Page 21, lines 4–7: “So far, in line with previous observations (Joechner et al., 2021), both analyses on SO-spindle co-occurrence suggested that sleep spindles in the adult-like fast frequency range rather than the more dominant, development-specific fast sleep spindles are coupled to SOs within all age groups.”.

Page 21, lines 10–13: “To capture the “maturational stage” of fast spindles for every participant, we resorted to our observed age-related differences in sleep spindles (Figure 1) and computed the distance between development-specific and adult-like fast sleep spindles within each individual at a given age.”

Regarding the suggestions to add robustness analyses, for the main results that show an association between sleep spindle but not slow oscillation maturity with slow oscillation spindle modulation strength, we have now added an analysis controlling for age (see also comment 5 above) as Supplementary file 1. Since these results hold and we also find similar results for the two different analyses approaches (partial least squares correlation and the generalized linear mixed-effects models) we are quite confident in this result for our sample.

Further, we also now included a control analysis to check whether our peri-event time histogram results might rely on differing numbers of events as Figure 4—figure supplement 2 (see comment 4). Based on this, and since we already use two different coupling measures that converge in their results and also with the literature (e.g., GarcíaPérez et al., 2022; Hahn et al., 2020), we believe that the slow oscillation-spindle coupling results are robust.

However, as also discussed on page 30, lines 20–24:

“Therefore, we cannot exclude a potentially important contribution of the development and characteristics of SOs for SO-spindle coupling. It may just be that the prominent age-related differences in fast sleep spindles across childhood and adolescence may be relatively more influential and/or that SO features other than the ones examined here may be effective in shaping SO-spindle coupling.”,

we cannot guarantee that our results generalize to other measures and samples. This remains to be tested in the future. As we close in the discussion, page 32, lines 1–4:

“Hence, our findings represent a promising starting point for future research addressing the precise relation between age-related changes in brain structure and function, the emergence of adult-like SO-spindle coupling, and cognitive development across childhood and adolescence.”

9. The time-frequency plots time-locked to SOs do not seem to show a meaningful comparison. This is usually done by comparing two conditions rather than to a baseline. I am not sure we can learn much from them, if there is just a huge cluster across the whole spectrum and time.

Thank you for sharing your concerns about our time frequency analyses. We understand, that seeing the huge clusters might evoke this concern (see also Reviewer 3’s suggestion on decreasing the α level). However, we did not compare our data to a baseline. To reiterate on our approach (see also “5.3.4.1. Time-frequency analyses”, page 52), we indeed did compare different conditions, namely trials with and trials without slow oscillations, on a first level (within participants) to extract maps showing in-/decreases in power during slow oscillation trials. The comparison against zero was done on a second level to establish significance of the *t*-maps of power differences across participants. Hence, the contrast in the time-frequency plots in Figure 3 (see above page 14) does reflect stable differences in wavelet power between trials with and without slow oscillations, which in our view is a meaningful contrast and has also been used by others (Bastian et al., 2022; Schreiner et al., 2022).

In order to narrow down the cluster, we tried the suggestion of Reviewer 3 and lowered our α level. However, this did not markedly change the size of the cluster (see Author response image 3). There just seems to be robustly higher power during slow oscillation trials in a broad frequency range compared to trials without slow oscillations.

**Author response image 3. sa2fig3:** Results of time frequency analyses with different cluster α. (**A**) Cluster α = 0.05 (two-sided, upper panels) and (**B**) cluster α = 0.01 (two-sided, lower panels). While the cluster shape changed slightly, there was no difference in the overall number of detected clusters.

The spatial, temporal, or frequency extent of clusters by itself is, however, not suited for inferences in the specificity of effects (Sassenhagen and Draschkow, 2019). The cluster size and location typically depend on several factors like the cluster forming procedure, filtering procedures, number of trials etc. Crucially, the exact shape of the cluster is itself typically not subject to statistical testing (Sassenhagen and Draschkow, 2019). Rather a cluster is inherently descriptive and includes with high chance the data points that reflect the true effect ,but at the same time also data points where the null hypothesis is true (Sassenhagen and Draschkow, 2019). Hence, a large cluster does not lessen our descriptive inference that sleep spindle power is enhanced during slow oscillation up peaks. Since the peri-event time histogram analyses converge with the time-frequency analyses we believe, we can be confident about the reported interpretation.

10. For better readability: please do not abbreviate spindles.Typos: "However, in neither age group, we did" (also: AFAIK "neither" is only used for two alternatives)

Thank you for this suggestion, we have changed this accordingly throughout the manuscript. Further, we have corrected our grammatical error, thank you for pointing this out.

Page 29, lines 10–12: “However, in none of the age groups did we observe strong evidence indicating that sleep spindles outside of the canonical fast spindle frequency range or at other topographic locations couple to distinct phases of the SO.”

Reviewer #2 (Recommendations for the authors):P3. Line 5: memories are not "transferred" to the cortex but strengthened in the cortex (since already encoded in the cortex).

Thank you for pointing out this error. We have replaced the term “transferred” to “strengthened”.

Page 3 lines 5–9: “Taken together, the complex wave sequence of SO up state and canonical fast sleep spindles, together with hippocampal activity, is considered to provide the scaffold for the precisely timed reactivation of initially fragile hippocampal memory representations and their strengthening in neocortical networks (Diekelmann and Born, 2010; Helfrich et al., 2019; Maingret et al., 2016; Staresina et al., 2015).”

After reading the whole article it is clear why the authors use "canonical" for the "normal" spindles, but it would be easier for readers if the authors the first time when they use the word, explain why and define it.

We thank the reviewer for making us aware that we have not defined explicitly enough what we mean by canonical fast sleep spindles. We agree that this is an important term for the presented research. Therefore, we already attempted to define canonical fast and slow spindles in the previous manuscript version by naming their frequency range and the topographical predominance in parentheses. In order to make clearer that we mean the fast spindle range that has been observed for young adults, we now updated the revised manuscript where appropriate, e.g.:

Page 2, lines 13–18: “Far from being an epiphenomenon, accumulating evidence suggests that the synchronization of canonical fast sleep spindles (i.e., spindles defined in young adults with a frequency of ≈ 12.5–16 Hz and a centro-parietal predominance; Cox et al., 2017), precisely during the up state of SOs provides an essential mechanism for neural communication, e.g., supporting systems memory consolidation during sleep (Hahn et al., 2020; Helfrich et al., 2018; Latchoumane et al., 2017; Mölle et al., 2002; Muehlroth et al., 2019)”

Page 2, line 23–page 3, line 3: “In addition to canonical fast sleep spindles, there is substantial evidence for a canonical slow sleep spindle type (i.e., spindles defined in young adults with a frequency of ≈ 9–12.5 Hz and a frontal predominance; Cox et al., 2017) in the human surface electroencephalogram (EEG; De Gennaro and Ferrara, 2003; Fernandez and Lüthi, 2020).”

Page 6, line 18–page 7, line 8: “Crucially, despite being faster, the frequency of the fast centroparietal spindles was specific to the age of the participants in a way that at younger ages, these dominant fast sleep spindles were not yet in the range of canonical fast spindles (i.e., as on average observed in adults with a frequency of ≈ 12.5–16 Hz; Cox et al., 2017; see Figure 1). For the vast majority of 14- to 18- and 20- to 26-year-olds, individually identified, dominant centro-parietal sleep spindles indeed matched the canonical fast sleep spindle frequency range (≈ 12.5–16 Hz, see Figure 1). For the majority of children though, dominant fast centro-parietal spindles were nested in the canonical slow spindle range (i.e., as on average observed in adults with a frequency of ≈ 9–12.5 Hz; Cox et a., 2017) and thus manifested in a development-specific fashion. Hence, the term “development-specific” will be employed to refer to individually determined, dominant fast centro-parietal sleep spindles across all age groups in the following. In contrast, since we also found dominant fast centro-parietal sleep spindles between 12.5–16 Hz in our adult sample, we will denote sleep spindle activity in this canonical fast spindle range and events detected exclusively within this canonical frequency range in centro-parietal sites as “adult-like” in our sample across all age groups.”

Page 8, lines 5–13: “After having characterized prevailing , development-specific fast centroparietal spindles across all our age groups, we were interested in how they differed from canonical fast spindles, i.e., those commonly, and also here, found in young adults with a frequency of ≈12.5–16 Hz (see Figure 1 for the present adult sample, cf. Cox et al., 2017; Ujma et al., 2015). Therefore, we additionally extracted fast spindles in centro-parietal electrodes by applying fixed frequency criteria between 12.5–16 Hz (henceforth, adult-like fast sleep spindles; see Figure 1—figure supplement 2 for an example raw EEG trace with development specific and adult-like fast spindles and Figure 1—figure supplement 1 for examples of averaged EEG signals time-locked to development-specific and adult-like fast sleep spindles).”

We hope that this helps improving clarity.

Reviewer #3 (Recommendations for the authors):Title: The title is rather unspecific with respect the age range and could be specified further.

Thank you for this suggestion, we have adapted the title accordingly to: “Sleep spindle maturity promotes slow oscillation-spindle coupling across child and adolescent development”

Figures: Almost all plots showing individual data points, means and raincloud plots are very hard to interpret and could need a small overhaul. In particular Figure 2, does not allow a visual assessment of the differences between cortical areas.

We agree that our plots were overcrowded with information and that too many plots were squeezed into too little space making it hard to interpret them. In the course of the revision, we have tried to reduce the amount of information within individual plots and to re-arrange them. Further, we rearranged the factors that are plotted in Figure 2 and hope that this improves the readability of the presented figures and increases their informative power.

Page 1, line 4: For consistency, the authors could also add the frequency range of SOs here.

Thank you for this suggestion, we have added this information in the abstract.

Page 1, lines 2–4: “The synchronization of canonical fast sleep spindle activity (12.5–16 Hz, adult-like) precisely during the slow oscillation (0.5–1 Hz) up peak is considered an essential feature of adult non-rapid eye movement sleep.”

Page 2, line 11: Here it is stated that how the coupling develops is an open question. However, at the end of the next paragraph, a previous publication is reported that shows how the coupling improves across childhood. Is this open question still valid then?Moreover, throughout the introduction, the authors repeat this open question several times with slights variations in the phrasing. I wonder if this is necessary or if the open question can be asked centrally before the final paragraph of the introduction.

We agree, that we have stated the same broad open question very extensively in the introduction. We have revised this and rephrased the question in the first (page 2, lines 11*–*12), second (page 3, lines 10*–*13), and third paragraph (page 4, lines 9*–*11) to:

1) “Yet, little is known about how this precise coalescence develops across childhood and adolescence.”

2) “However, the precise coupling of sleep spindle activity to SOs, described above, does not seem to be fully present and functional from early childhood on. Recent evidence in rodents and humans indicates that the temporal co-ordination of SOs and spindles improves across childhood and adolescence (García-Pérez et al., 2022; Hahn et al., 2020; Joechner et al., 2021).”

3) “Nevertheless, it is still unclear how developments in sleep spindles and SOs interact to promote precise, adult-like temporal synchronization of sleep spindles during SOs across childhood and adolescence.”

Even though we have rephrased the original question in the second paragraph (page 3, lines 10*–*13) to a declarative sentence (see (2)) and the existence of some excellent literature, we are convinced, that the very broad question on how slow oscillation-spindle coupling develops is still valid. We are at the moment aware of only a handful of papers that examine slow oscillation-spindle coupling in a developmental context (e.g., García-Pérez et al., 2022; Hahn et al., 2020, 2022; Joechner et al., 2021; Kurz et al., 2021; Piantoni et al., 2013). Amongst these, most publications focus on late childhood and adolescence. While the mentioned research provides important insights into slow oscillation-spindle coupling development, a lot of open, more specific, questions remain e.g., how is coupling expressed in young childhood and infancy? What drives developments in slow oscillation coupling across maturation? How is this connected to memory consolidation?

We hope that our adjustments removed redundancies and improve introducing our narrow question on how sleep spindle and slow oscillation developments relate to developments in their coupling (see (3) and page 4, lines 9*–*11).

Page 6, line 4: Please see my comment in the public review. I think the assessment of average frequency of spindle events is redundant or even circular as they are based on the power spectra.

We thank Reviewer 3 for highlighting redundancies in our reported results. We agree that reporting the results of both the peak frequencies and the average frequencies of the individually identified sleep spindles is unnecessary.

However, we have several reasons for assessing the average frequency of sleep spindle events:

First, we frequently encounter the problem that detection based on adult-like criteria can lead to the identification of sleep spindles below the lower boundary of this window; specifically, for young children whose peak frequencies are markedly lower. This leads to the phenomenon that average spindle frequencies can be lower (theoretically also higher) than the upper and lower boundaries of the intended search frequency window if these powerful events are included and dominate. Hence, assessing the average frequency of adult-like fast sleep spindles is in our view important to validate our methodical approach and to ensure that we detected sleep spindles in the range of interest in all age groups.

Second, we consider the comparison of the frequencies of spindles detected based on individual peak frequencies and fixed adult-like criteria central to quantify sleep spindle development. Given the absence of spectral peaks for adult-like spindles, the average frequencies of detected events for both sleep spindle types represents a comparable measure in both spindle types.

Therefore, we have now decided to concentrate on reporting the average frequencies of spindle events in the main text and to only report the statistical results from the peak frequency comparisons in Supplementary file 1 for completeness.

Page 6, line 18: The introduction of development-specific feels unnecessary and to some degree arbitrary. At least for me, it only led to overly complicated/cluttered sentences. Is this term really required or can it be omitted while the main message is conveyed properly.

Thank you for sharing your concerns with the introduced terminology. We agree that in our attempt to be maximally descriptive and discriminative in our terminology some long terms have emerged. However, in our view the term “development-specific” conveys a central message: despite between-person differences, dominant fast centro-parietal sleep spindle activity manifest very differently across (lifespan) development (cf. Cox et al., 2017; Muehlroth and Werkle‐Bergner, 2020). Hence, on a more general scale, this term does not only apply to the phenomenon that we find slower fast sleep spindles in young children in the context of this paper but is also applicable to different time scales and contexts and hopefully fosters usage of adaptive methods to capture a variable phenomenon.

We consider the contrast of dominant sleep spindles being manifested in a development specific fashion across maturation and yet a stable presence of slow oscillation-spindle coupling for spindles in the adult-like fast spindle range one of the main conundrums. Hence, to highlight the fact of a low presence of coupled adult-like fast sleep spindles in the face of distinctly manifested dominant sleep spindles, we would like to stick to the term “development-specific” in the context of this paper in cases where we directly contrast results based on the different sleep spindle detection approaches. We believe that using these descriptive and discriminative terms helps the reader to better follow the analysis steps.

However, we agree that these terms are very long. Therefore, we went through the manuscript in the course of this revision and tried to reduce its usage in all sections except the Results section. We hope this makes sentences in these parts of the manuscript less cluttered.

Page 10, Figure 1: It would help the reader if the age ranges are illustrated in subpanel A. Moreover, as stated above, symbols for the different spindle types and the marginal means are very hard to distinguish. Also, I was at first not aware that the adult-like spindles even had a rain cloud. Perhaps the authors can rearrange/resize the figure.Related to subpanel A, its known that adults sometimes do not show slow spindle peaks or shows both a slow and fast spindle peak at one electrode (e.g., Mölle et al., 2011). At least the former seems to be the same here. How did the authors proceed if that was the case? I was wondering if it would be more informative to sort the individual subjects peak frequency per age group.

We thank Reviewer 3 for his suggestions to improve our figures, we have tried to implement them and hope this makes them easier to read and more informative (see comment on page 24 above). Regarding the suggestion for Figure 1 (see page 25 of this response letter for the revised Figure 1), we agree that it is most informative to sort participants by age groups. Indeed, this is what we already did. As indicated by in the figure text, page 12, line 4: “Data are ordered by age from bottom to top (y-axis)”, we did sort the power spectral plots not only by age group but also within age group by age (in months).

Regarding the non-trivial question on the issue of missing or several peaks in the sleep spindle frequency range for the identification of slow and fast sleep spindles, we consider this one of the core problems when differentiating slow and fast spindles based on predefined frequency criteria or across averaged topographies (Cox et al., 2017; Joechner et al., 2021). This is why we have neither identified peaks nor spindle events as slow or fast based on their absolute frequency but we took a slightly different approach that we want to reiterate and elaborate shortly in the following.

As also described on pages 50 /51 under “5.3.3 Sleep spindle and slow oscillation detection”, we implemented the identification for spectral peaks between 9 and 16 Hz separately for averaged frontal and centro-parietal electrodes, where usually slow and fast sleep spindles dominate, respectively, and based on the background-corrected power spectra. We then identified the maximum peak between 9 and 16 Hz, that also had a zero crossing in the 1^st^ derivative, as the dominant peak frequency. Hence, in case of the presence of a less prominent second (third…) peak, this peak (s) was/were not informing the peak frequency measure. However, since we used the detected frontal and centro-parietal spectral peaks as center frequencies of a quite broad frequency window (± 1.5 Hz) for spindle event detection in frontal or centro-parietal electrodes, spindles at slightly different frequencies than the peak frequency were also detected.

Importantly, in every participant, we found one prominent peak in frontal and centro-parietal background corrected power spectra. We controlled peak detection by our algorithm through visual inspection of all power spectra and in all cases the algorithm captured the prominent peak per topography and individual. Further, only a rare number of participants showed a second, less prominent and flatter peak in averaged frontal or centro-parietal power spectra, that was in close proximity to the more prominent peak.

Crucially, at this stage, we do not yet classify peaks or spindles detected based on the peak frequencies as slow or fast based on their frequency (so below or above a certain frequency threshold) as, to the best of our knowledge, done in Mölle et al. (2011), or on topography alone. Therefore, we did not run into the problem of missing slow or fast peaks.

Rather, based on the assumption that slow(er) spindles usually dominate in frontal and fast(er) spindles in centro-parietal areas, we then tested whether spectral peak frequencies and frontal spindles detected based on the peak frequency would have a slower frequency than spectral peaks and centro-parietal spindles, that were extracted on the individual peak frequencies. Since this was statistically evident, the results imply to us that, based on both topography and frequency (Anderer et al., 2001; Cox et al., 2017; Gibbs and Gibbs, 1950; Mölle et al., 2011), we find evidence for slow frontal and fast-centro-parietal sleep spindles. This does not automatically mean that either frontal or centro-parietal spindles are in the canonical slow (9*–*12.5 Hz) or fast spindle range (12.5*–*16 Hz Hz) for every participant. While we explicitly focus on this feature for centro-parietal spindles in the context of maturation, the inverse phenomenon (a slower frontal spindle peak that is faster than the canonical slow spindle peak) could also apply.

Taken together, by detecting peaks in background corrected power spectra separately for frontal and centro-parietal electrodes, we increased the chance of detecting prominent peaks that reflect true rhythmic neural activity. While our results imply that these peaks and the derived sleep spindles were on average slower in frontal than centro-parietal derivations, this does not mean that either these slow frontal or centro-parietal peaks or sleep spindle events would be considered as slow or fast based on canonical definitions.

Page 14, line 4: What do the authors mean by "more precise"? The size of the cluster or its location with respect to the SO up-state? Is it really possible to make conclusion about the precision using a TFR-based analysis affected by temporal and frequency smearing?

In general, we mean with “more precise” the temporal clustering closer to the slow oscillation up peak on a descriptive level (time-frequency and peri-event time histogram (PETH) analyses). We are aware that clusters from cluster-based permutation tests do not justify statistical inferences about the spatio-temporal extent or location of clusters, while on a descriptive level, description of effect shapes and locations is justified (Sassenhagen and Draschkow, 2019). Further, in the case of our time-frequency analyses, just like in every time-frequency analysis, temporal and frequency smearing limit exact interpretability of frequency and temporal boundaries (Cohen, 2019). We have chosen a 12-cycle wavelet which rather emphasizes good frequency resolution over temporal precision. However, while this limits identification of exact time and frequency differences of effects, it allows careful description of the effects. Therefore, we already have tried to be careful in our description of the time-frequency analyses by using terms as “seemed stronger and more precise” (actually now: " seemed to be stronger and more precise during the SO up peak”) on page 16, line 16 and highlighting that we are on a “descriptive level” (page 16 line 12). The same applies to the description of the PETH results, where the concerns regarding temporal or frequency smearing do not apply. Since both analyses converge in terms of descriptive interpretation of the temporal clustering of increased and decreased sleep spindle occurrence we feel confident in describing our results in terms of precision.

In order to resolve issues with our use of the term “more precise” and to avoid misleading the readers, we revised the respective parts in the manuscript to be more explicit, e.g.:

Page 26, lines 13–16: “Surprisingly, the inspection of SO-spindle coupling patterns suggested synchronization being driven by spindles in the adult-like, canonical fast spindle range—even in the younger age groups but notably less precisely timed during the SO cycle.”

Page 27, lines12/13: “Overall, our findings suggest that the temporally precise synchronization of sleep spindles during the up peak of SOs might not be an inherent feature of the thalamocortical system.”

Page 29, lines 16–20: “Surprisingly though, the modulation of adult-like fast sleep spindles during SOs was well detectable across all age groups with this co-occurrence pattern becoming more precisely timed with SO up and down peaks with older age—despite huge age-related differences in their occurrence rate and the fact that adult-like fast sleep spindles showed very low density in both child cohorts (see Figure 1).” Page 14, Figure 2.

A comparison between SO vs. SO-free intervals includes a strong difference in overall spectral power, hence the wide significant cluster. The authors could consider decreasing their α-value further to tease out clusters more clearly. Furthermore, the dashed window highlighting the spindle range is not really visible.

Page 14, Figure 2. A comparison between SO vs. SO-free intervals includes a strong difference in overall spectral power, hence the wide significant cluster. The authors could consider decreasing their α-value further to tease out clusters more clearly. Furthermore, the dashed window highlighting the spindle range is not really visible.

Thank you for this suggestion. We have tried decreasing the α level for the cluster-based permutation test on the between-participant level, however, the cluster sizes did not change considerably (Author response image 3). Therefore, we decided to stay with the original α.

Concerning the figures on our time-frequency results, we have now intensified the color of the window indicating the spindle frequency range and hope this improves our figures (see Figure 3).

Page 16, Figure 4: Again, this is reflecting my personal opinion, but this the central result and it somehow feels like it does not get enough attention. Anyhow, what does the horizontal jiggly line represent? It is also stated that the empirical PETHs are compared to a surrogate distribution, however, is that really necessary? The authors have SO-free intervals that could be used for a cluster-based permutation test as well.

The horizontal jiggly line in the peri-event time histograms (PETH) reflects the surrogate distribution that was the comparison condition to establish statistical significance. We describe the line in the figure legend with the following sentence:

“Green asterisks mark increased spindle occurrence (positive cluster, cluster-based permutation test, cluster α <.05, two-sided test) and red asterisks mark decreased spindle occurrence (negative cluster, cluster-based permutation test, cluster α <.05, two-sided test) compared to random occurrence (horizontal line)”.

We now added more explicitly that the horizontal line represents the surrogate comparison distribution and hope this increases interpretation:

Page 19, lines 4–7: “Green asterisks mark increased spindle occurrence (positive cluster, cluster-based permutation test, cluster α <.05, two-sided test) and red asterisks mark decreased spindle occurrence (negative cluster, cluster-based permutation test, cluster α < .05, two-sided test) compared to a surrogate distribution representing random occurrence (horizontal line).”

Concerning the choice of reference distribution to statistically determine whether spindle occurrence during slow oscillation windows was temporally clustered, we consider randomly shuffling the original data a straight forward, easy procedure to obtain a comparison distribution that misses temporal structure but otherwise contains the same properties as the original distribution (Lancaster et al., 2018; for similar approaches, see Bastian et al., 2022; Muehlroth et al., 2019). Therefore, it allows to easily determine whether there is a temporal structure in the data. We agree, that alternatively, we could have also employed slow oscillation-free trials to construct comparison distributions that would be suitable to test whether a temporal structure in the PETHs would be specific to slow oscillation trials.

Lastly, since we agree that these are important results, we have changed the headings on page 16 to:

“2.3 Temporal modulation of adult-like fast sleep spindle power during slow oscillations is present across all ages” and on page 17 to “2.4 At older age, development specific fast sleep spindle occurrence is more strongly modulated by slow oscillations while slow oscillation-adult-like fast spindle coupling becomes temporally more precise”.

We hope this helps highlighting the results.

Page 19, Figure 6: Can the author please explain what the main message of subpanel A is? This somehow eludes me. Also, how come there is not separation into the different age groups?

Thank you for making us aware of the lack of clarity of our description and plots on this part of the partial least squares correlation (PLSC). As described under “5.4.2. Association between sleep spindle and slow oscillation maturity with slow oscillation-spindle coupling”, page 55/56, we determined the reliability of the sleep spindle maturity profile, that was associated with a specific pattern of power modulations during slow oscillations by means of:

“the correlation between the latent time-frequency variable and the three raw spindle maturity variables. These values are comparable to the spindle maturity weights and indicate whether the association between the time-frequency pattern and individual spindle maturity variables is in the same or different direction (McIntosh and Lobaugh, 2004). Stability of these correlation patterns was determined by their bootstrap estimated 95% confidence intervals (McIntosh and Lobaugh, 2004).”

Figure 6 displays the pattern of how the individual fast sleep spindle maturity features are related to the corresponding coupling pattern (Figure 6). In this case all features were reliably and in the same way related to the slow oscillation-spindle coupling pattern. We have tried to improve the clarity of Figure 6 by revising its explanation in the main text.

Further, there is no separation by age group since we considered all age groups within one PLS analysis as noted on page 55 under “5.4.2. Association between sleep spindle and slow oscillation maturity with slow oscillation-spindle coupling”. Hence Figure 6 reflect the associated pattern of sleep spindle maturity and slow oscillation-spindle coupling across all age groups. We are aware that this means that there are participants with differing dependencies in the analyses. However, this only acts against us in terms of power since the participants from the longitudinal cohort are treated as independent individuals. While we had this information described in the method section, we admittedly did not mention this clearly in the Results section. We have now added this information also in the Results section on page 23, lines 11-24:

“For one, we examined the relation between the pattern of power modulations in the spindle frequency range (9–16 Hz) during the complete SO trial (down peak ± 1.2 sec, cf. time frequency t-maps, Figure 3) with the fast spindle maturity scores for sleep spindle frequency, density, and amplitude using a partial least squares correlation (PLSC; Krishnan et al., 2011) across all participants. This analysis provides pairs of SO-spindle coupling profiles and associated, multivariate patterns of spindle maturity scores (spindle maturity profile). Based on a permutation test, we identified one significant pair of a spindle maturity profile (Figure 6) and a specific time-frequency SO-spindle coupling pattern (Figure 6; singular vector pair p <.001). All fast spindle maturity measures contributed reliably and positively to this significant, positive correlation (as indicated by the direction of values in the spindle maturity profile and the non-zero crossing confidence intervals in Figure 6). As can be inferred from Figure 6, the SO-spindle coupling pattern suggests, that higher fast sleep spindle maturity in all features (Figure 6) was associated with a more adult-like SO-spindle coupling pattern, reflected in: […]”

References

Anderer, P., Klösch, G., Gruber, G., Trenker, E., Pascual-Marqui, R. D., Zeitlhofer, J., Barbanoj, M. J., Rappelsberger, P., and Saletu, B. (2001). Low-resolution brain electromagnetic tomography revealed simultaneously active frontal and parietal sleep spindle sources in the human cortex. Neuroscience, 103(3), 581–592. https://doi.org/10.1016/S0306-4522(01)00028-8

Aru, J., Aru, J., Priesemann, V., Wibral, M., Lana, L., Pipa, G., Singer, W., and Vicente, R. (2015). Untangling cross-frequency coupling in neuroscience. Current Opinion in Neurobiology, 31, 51–61. https://doi.org/10.1016/j.conb.2014.08.002

Attarian, H., and Viola-Saltzman, M. (Hrsg.). (2020). Sleep Disorders in Women: A Guide to Practical Management. Springer International Publishing. https://doi.org/10.1007/978-3-030-40842-8

Barakat, M., Doyon, J., Debas, K., Vandewalle, G., Morin, A., Poirier, G., Martin, N., Lafortune, M., Karni, A., Ungerleider, L. G., Benali, H., and Carrier, J. (2011). Fast and slow spindle involvement in the consolidation of a new motor sequence. Behavioural Brain Research, 217(1), 117–121. https://doi.org/10.1016/j.bbr.2010.10.019

Bastian, L., Samanta, A., Ribeiro de Paula, D., Weber, F. D., Schoenfeld, R., Dresler, M., and Genzel, L. (2022). Spindle–slow oscillation coupling correlates with memory performance and connectivity changes in a hippocampal network after sleep. Human Brain Mapping, 43(13), 3923–3943. https://doi.org/10.1002/hbm.25893

Bates, D., Mächler, M., Bolker, B., and Walker, S. (2015). Fitting linear mixed-effects models using lme4. Journal of Statistical Software, 67(1), 1–48. https://doi.org/10.18637/jss.v067.i01

Campbell, I. G., and Feinberg, I. (2016). Maturational patterns of σ frequency power across childhood and adolescence: A longitudinal study. Sleep, 39(1), 193–201. https://doi.org/10.5665/sleep.5346

Cohen, M. X. (2019). A better way to define and describe Morlet wavelets for time-frequency analysis. NeuroImage, 199, 81–86. https://doi.org/10.1016/j.neuroimage.2019.05.048

Cowan, E., Liu, A., Henin, S., Kothare, S., Devinsky, O., and Davachi, L. (2020). Sleep spindles promote the restructuring of memory representations in ventromedial prefrontal cortex through enhanced hippocampal-cortical functional connectivity. Journal of Neuroscience, 40(9), 1909–1919. https://doi.org/10.1523/JNEUROSCI.1946-19.2020

Cox, R., Schapiro, A. C., Manoach, D. S., and Stickgold, R. (2017). Individual differences in frequency and topography of slow and fast sleep spindles. Frontiers in Human Neuroscience, 11, Article 433. https://doi.org/10.3389/fnhum.2017.00433

D’Atri, A., Novelli, L., Ferrara, M., Bruni, O., and De Gennaro, L. (2018). Different maturational changes of fast and slow sleep spindles in the first four years of life. Sleep Medicine, 42, 73–82. https://doi.org/10.1016/j.sleep.2017.11.1138

De Gennaro, L., and Ferrara, M. (2003). Sleep spindles: An overview. Sleep Medicine Reviews, 7(5), 423–440. https://doi.org/10.1053/smrv.2002.0252

Diekelmann, S., and Born, J. (2010). The memory function of sleep. Nature Reviews Neuroscience, 11(2), 114–126. https://doi.org/10.1038/nrn2762

Fernandez, L. M. J., and Lüthi, A. (2020). Sleep spindles: Mechanisms and functions. Physiological Reviews, 100(2), 805–868. https://doi.org/10.1152/physrev.00042.2018

Franco, P., Putois, B., Guyon, A., Raoux, A., Papadopoulou, M., Guignard-Perret, A., Bat-Pitault, F., Hartley, S., and Plancoulaine, S. (2020). Sleep during development: Sex and gender differences. Sleep Medicine Reviews, 51, 101276. https://doi.org/10.1016/j.smrv.2020.101276

García-Pérez, M. A., Irani, M., Tiznado, V., Bustamante, T., Inostroza, M., Maldonado, P. E., and Valdés, J. L. (2022). Cortico-Hippocampal Oscillations Are Associated With the Developmental Onset of Hippocampal-Dependent Memory. Frontiers in Neuroscience, 16, 891523. https://doi.org/10.3389/fnins.2022.891523

Gibbs F.A., and Gibbs E.L. (1950). Atlas of Electroencephalography. Reading, MA.: Addison‐Wesley, 1950.

Hahn, M. A., Bothe, K., Heib, D., Schabus, M., Helfrich, R. F., and Hoedlmoser, K. (2022). Slow oscillation– spindle coupling strength predicts real-life gross-motor learning in adolescents and adults. *eLife*, 11, Article e66761. https://doi.org/10.7554/eLife.66761

Hahn, M. A., Heib, D., Schabus, M., Hoedlmoser, K., and Helfrich, R. F. (2020). Slow oscillation-spindle coupling predicts enhanced memory formation from childhood to adolescence. *eLife*, 9, Article e53730. https://doi.org/10.7554/eLife.53730

Hahn, M. A., Joechner, A.-K., Roell, J., Schabus, M., Heib, D. P., Gruber, G., Peigneux, P., and Hoedlmoser, K. (2019). Developmental changes of sleep spindles and their impact on sleep‐dependent memory consolidation and general cognitive abilities: A longitudinal approach. Developmental Science, 22(1), Article e12706. https://doi.org/10.1111/desc.12706

Helfrich, R. F., Lendner, J. D., Mander, B. A., Guillen, H., Paff, M., Mnatsakanyan, L., Vadera, S., Walker, M. P., Lin, J. J., and Knight, R. T. (2019). Bidirectional prefrontal-hippocampal dynamics organize information transfer during sleep in humans. Nature Communications, 10(1), Article 3572. https://doi.org/10.1038/s41467-019-11444-x

Helfrich, R. F., Mander, B. A., Jagust, W. J., Knight, R. T., and Walker, M. P. (2018). Old brains come uncoupled in sleep: Slow wave-spindle synchrony, brain atrophy, and forgetting. Neuron, 97(1), 221-230.e4. https://doi.org/10.1016/j.neuron.2017.11.020

Joechner, A.-K., Wehmeier, S., and Werkle-Bergner, M. (2021). Electrophysiological indicators of sleepassociated memory consolidation in 5‐ to 6-year-old children. Psychophysiology, 58(8), Article e13829. https://doi.org/10.1111/psyp.13829

Krishnan, A., Williams, L. J., McIntosh, A. R., and Abdi, H. (2011). Partial Least Squares (PLS) methods for neuroimaging: A tutorial and review. NeuroImage, 56(2), 455–475. https://doi.org/10.1016/j.neuroimage.2010.07.034

Kurz, E.-M., Conzelmann, A., Barth, G. M., Renner, T. J., Zinke, K., and Born, J. (2021). How do children with autism spectrum disorder form gist memory during sleep? A study of slow oscillation–spindle coupling. Sleep, 44(6), Article zsaa290. https://doi.org/10.1093/sleep/zsaa290

Kuznetsova, A., Brockhoff, P. B., and Christensen, R. H. B. (2017). lmerTest Package: Tests in linear mixed effects models. Journal of Statistical Software, 82(13). https://doi.org/10.18637/jss.v082.i13

Lancaster, G., Iatsenko, D., Pidde, A., Ticcinelli, V., and Stefanovska, A. (2018). Surrogate data for hypothesis testing of physical systems. Physics Reports, 748, 1–60. https://doi.org/10.1016/j.physrep.2018.06.001

Latchoumane, C.-F. V., Ngo, H.-V. V., Born, J., and Shin, H.-S. (2017). Thalamic spindles promote memory formation during sleep through triple phase-locking of cortical, thalamic, and hippocampal rhythms. Neuron, 95(2), 424-435.e6. https://doi.org/10.1016/j.neuron.2017.06.025

Lindenberger, U., and Pötter, U. (1998). The complex nature of unique and shared effects in hierarchical linear regression: Implications for developmental psychology. Psychological Methods, 3(2), 218– 230.

Lindenberger, U., von Oertzen, T., Ghisletta, P., and Hertzog, C. (2011). Cross-sectional age variance extraction: What’s change got to do with it? Psychology and Aging, 26(1), 34–47. https://doi.org/10.1037/a0020525

Maingret, N., Girardeau, G., Todorova, R., Goutierre, M., and Zugaro, M. (2016). Hippocampo-cortical coupling mediates memory consolidation during sleep. Nature Neuroscience, 19(7), 959–964. https://doi.org/10.1038/nn.4304

McIntosh, A. R., and Lobaugh, N. J. (2004). Partial least squares analysis of neuroimaging data: Applications and advances. NeuroImage, 23, S250–S263. https://doi.org/10.1016/j.neuroimage.2004.07.020

Mölle, M., Bergmann, T. O., Marshall, L., and Born, J. (2011). Fast and slow spindles during the sleep slow oscillation: Disparate coalescence and engagement in memory processing. Sleep, 34(10), 1411– 1421. https://doi.org/10.5665/SLEEP.1290

Mölle, M., Marshall, L., Gais, S., and Born, J. (2002). Grouping of spindle activity during slow oscillations in human non-rapid eye movement sleep. The Journal of Neuroscience, 22(24), 10941–10947. https://doi.org/10.1523/JNEUROSCI.22-24-10941.2002

Mong, J. A., and Cusmano, D. M. (2016). Sex differences in sleep: Impact of biological sex and sex steroids. Philosophical Transactions of the Royal Society B: Biological Sciences, 371(1688), 20150110. https://doi.org/10.1098/rstb.2015.0110

Muehlroth, B. E., Sander, M. C., Fandakova, Y., Grandy, T. H., Rasch, B., Shing, Y. L., and Werkle-Bergner, M. (2019). Precise slow oscillation–spindle coupling promotes memory consolidation in younger and older adults. Scientific Reports, 9(1), Article 1940. https://doi.org/10.1038/s41598-01836557-z

Muehlroth, B. E., and Werkle‐Bergner, M. (2020). Understanding the interplay of sleep and aging: Methodological challenges. Psychophysiology, 57(3), Article e13523. https://doi.org/10.1111/psyp.13523

Nicolas, A., Petit, D., Rompré, S., and Montplaisir, J. (2001). Sleep spindle characteristics in healthy subjects of different age groups. Clinical Neurophysiology, 112(3), 521–527. https://doi.org/10.1016/S1388-2457(00)00556-3

Niknazar, M., Krishnan, G. P., Bazhenov, M., and Mednick, S. C. (2015). Coupling of thalamocortical sleep oscillations are important for memory consolidation in humans. PLoS ONE, 10(12), e0144720. https://doi.org/10.1371/journal.pone.0144720

Olbrich, E., Rusterholz, T., LeBourgeois, M. K., and Achermann, P. (2017). Developmental changes in sleep oscillations during early childhood. Neural Plasticity, 2017, Article ID 6160959. https://doi.org/10.1155/2017/6160959

Peiffer, A., Brichet, M., De Tiège, X., Peigneux, P., and Urbain, C. (2020). The power of children’s sleep: Improved declarative memory consolidation in children compared with adults. Scientific Reports,

10(1), Article 9979. https://doi.org/10.1038/s41598-020-66880-3

Piantoni, G., Astill, R. G., Raymann, R. J. E. M., Vis, J. C., Coppens, J. E., and Van Someren, E. J. W. (2013). Modulation of γ and spindle-range power by slow oscillations in scalp sleep EEG of children. International Journal of Psychophysiology, 89(2), 252–258. https://doi.org/10.1016/j.ijpsycho.2013.01.017

Purcell, S. M., Manoach, D. S., Demanuele, C., Cade, B. E., Mariani, S., Cox, R., Panagiotaropoulou, G., Saxena, R., Pan, J. Q., Smoller, J. W., Redline, S., and Stickgold, R. (2017). Characterizing sleep spindles in 11,630 individuals from the National Sleep Research Resource. Nature Communications, 8(1), Article 15930. https://doi.org/10.1038/ncomms15930

Rasch, B., and Born, J. (2013). About sleep’s role in memory. Physiological Reviews, 93(2), 681–766. https://doi.org/10.1152/physrev.00032.2012

Sassenhagen, J., and Draschkow, D. (2019). Cluster‐based permutation tests of MEG/EEG data do not establish significance of effect latency or location. Psychophysiology, 56(6), e13335. https://doi.org/10.1111/psyp.13335

Scholle, S., Zwacka, G., and Scholle, H. C. (2007). Sleep spindle evolution from infancy to adolescence. Clinical Neurophysiology, 118(7), 1525–1531. https://doi.org/10.1016/j.clinph.2007.03.007

Schreiner, T., Kaufmann, E., Noachtar, S., Mehrkens, J.-H., and Staudigl, T. (2022). The human thalamus orchestrates neocortical oscillations during NREM sleep. Nature Communications, 13(1), 5231. https://doi.org/10.1038/s41467-022-32840-w

Shinomiya, S., Nagata, K., Takahashi, K., and Masumura, T. (1999). Development of sleep spindles in young children and adolescents. Clinical Electroencephalography, 30(2), 39–43. https://doi.org/10.1177/155005949903000203

Staresina, B. P., Bergmann, T. O., Bonnefond, M., van der Meij, R., Jensen, O., Deuker, L., Elger, C. E., Axmacher, N., and Fell, J. (2015). Hierarchical nesting of slow oscillations, spindles and ripples in the human hippocampus during sleep. Nature Neuroscience, 18(11), 1679–1686. https://doi.org/10.1038/nn.4119

Ujma, P. P., Gombos, F., Genzel, L., Konrad, B. N., Simor, P., Steiger, A., Dresler, M., and Bódizs, R. (2015). A comparison of two sleep spindle detection methods based on all night averages: Individually adjusted vs. fixed frequencies. Frontiers in Human Neuroscience, 9, Article 52. https://doi.org/10.3389/fnhum.2015.00052

Urbain, C., De Tiège, X., Op De Beeck, M., Bourguignon, M., Wens, V., Verheulpen, D., Van Bogaert, P., and Peigneux, P. (2016). Sleep in children triggers rapid reorganization of memory-related brain processes. NeuroImage, 134, 213–222. https://doi.org/10.1016/j.neuroimage.2016.03.055

Wilhelm, I., Prehn-Kristensen, A., and Born, J. (2012). Sleep-dependent memory consolidation – What can be learnt from children? Neuroscience and Biobehavioral Reviews, 36(7), 1718–1728. https://doi.org/10.1016/j.neubiorev.2012.03.002

Wilhelm, I., Rose, M., Imhof, K. I., Rasch, B., Büchel, C., and Born, J. (2013). The sleeping child outplays the adult’s capacity to convert implicit into explicit knowledge. Nature Neuroscience, 16(4), 391–393. https://doi.org/10.1038/nn.3343

Zhang, Z. Y., Campbell, I. G., Dhayagude, P., Espino, H. C., and Feinberg, I. (2021). Longitudinal analysis of sleep spindle maturation from childhood through late adolescence. Journal of Neuroscience, 41(19), 4253–4261. https://doi.org/10.1523/JNEUROSCI.2370-20.2021